# Past, Present, and Future of Copper Mine Tailings Governance in Chile (1905–2022): A Review in One of the Leading Mining Countries in the World

**DOI:** 10.3390/ijerph192013060

**Published:** 2022-10-11

**Authors:** Carlos Cacciuttolo, Edison Atencio

**Affiliations:** 1Civil Works and Geology Department, Catholic University of Temuco, Temuco 4780000, Chile; 2School of Civil Engineering, Pontificia Universidad Católica de Valparaíso, Av. Brasil 2147, Valparaíso 2340000, Chile; 3Department of Management, Economics, and Industrial Engineering, Politecnico di Milano, Via Lambruschini 4b, Bovisa, 20156 Milan, Italy

**Keywords:** mine tailings, tailings disposal, tailings reprocessing, rare earth elements (REEs), submarine deep-sea tailings disposal, tailings governance, circular economy

## Abstract

How mine tailings storage facilities (TSF) are managed reflects the history, regulatory framework, and environment of a country and locale of the mine. Despite many attempts to find an environmentally friendly strategy for tailings management and governance that balances the needs of society and the ecosystem, there is no worldwide agreement regarding the best practices for tailings management and governance. This article reviews the evolution of copper tailings management and governance in Chile, current practices, and changes that could be or may need to be made to improve practices in response to local environmental conditions and local tolerance for risk. The progress to date in developing a holistic tailings management strategy is summarized. This article also describes recent proposals for the best available technologies (BATs), case histories of Chilean TSF using conventional technology, thickened tailings, paste tailings, filtered tailings, water use reduction, tailings reprocessing to obtain rare earth elements (REEs), circular economy, submarine deep-sea tailings disposal, and ways to avoid failure in a seismic region. Finally, the Chilean tailings industry’s pending issues and future challenges in reducing the socioenvironmental impacts of tailings are presented, including advances made and lessons learned in developing more environmentally friendly solutions.

## 1. Introduction

Considering the history and evolution of copper oxide ore deposits, it is known that these became increasingly scarce and could not meet the increasing demand for metals during industrialization. This brought, as a consequence, the beginning of the exploration of deposits of copper sulphide minerals. The metallurgical technique called flotation began to be implemented in mining projects in the late nineteenth and early twentieth centuries (1860–1905), which allowed the separation of metal sulphide minerals from gangue mineralogy or copper mine tailings. The flotation, also called physical–chemical concentration of minerals, allowed to selectively float a mineral of economic interest (e.g., chalcopyrite), making the surface of the mineral hydrophobic, which then floats by introducing air bubbles and generating copper mine tailings. Other sulphide minerals, such as pyrite (FeS_2_), without economic interest, can be eliminated from flotation by adjusting the pH (e.g., alkaline circuit). Pyrite is the most frequent complementary mineral in sulphide ore deposits [1].

Due to the selective properties of the metallurgical flotation process, the unprofitable part, which represents, in the case of copper minerals, about 97–99% of the ore treated, is classified as residue or gangue and is called “mine tailings”. In general, copper mine tailings are the waste material that remains after the economic fraction is extracted from the mineral ore. Copper mine tailings consist of a slurry of ground rock (considering different metals), water, and chemical reagents that remain after metallurgical processing. The composition of mine tailings varies according to the mineralogy of the ore deposit and how the ore is processed [2]. These mine tailings are generally made up of solids and water, they are transported in pipes or channels to a tailings storage facility usually built with a reservoir and dam. Formerly, in some cases, tailings were discharged into rivers and the sea; today, this is prohibited, and currently tailings are stored in valleys or flat areas. It is known that mine tailings do not have economic concentrations of metals, but they still represent a strong enrichment of these metals in relation to the earth’s crust. Additionally, according to geochemistry, it is known that tailings contain other sulphide minerals, such as pyrite (FeS_2_), arsenopyrite (FeAsS), enargite (Cu_3_AsS_4_), galena (PbS), and sphalerite (ZnS), which can generate a future uncontrolled release of metals which results in an acid rock drainage (ARD) [3]. Modern tailings deposits from Cu mines average about 0.1 wt.% Cu, while older tailings storage facilities may still contain concentrations between 0.2 and 0.6 wt.% Cu [1].

Within the mining world, it is known that Chile has a reputation as a mining country with a long and proud history of mining. The historical mining activity of Chile includes the extraction and processing of the following valuable metals: copper, iron, gold, silver, and molybdenum.

Chile had an important mining boom carried out by local entrepreneurs in the mid-19th century, considering the extraction of high-grade minerals, both copper and silver [4]. However, it is important to mention that all the high-grade mines ceased operations by the end of the 19th century. Therefore, the exploitation of lower grade minerals required new mineral processing technology and more investment capital, neither of which, at the beginning of the 20th century, were available in Chile [4]. Large foreign mining companies entered the copper industry in Chile at the beginning of the 20th century, such as the case of the El Teniente Mine near the city of Rancagua, in which William Braden became involved in 1905. Later, over the years, the following foreign mining companies began operations: Chile Exploration Company (Chuquicamata, 1912, Figure 1) and Minería del Cobre de los Andes (Potrerillos, 1916) [5].

Today Chile has one of the most productive and vibrant copper geological, mining and metallurgical industries in the world [7]. As ore production from its mines increased, so have mine tailings storage facilities. Chile has a diversity of relief or topography, also of climates in its different regions from north to south, and the risk of earthquakes is an ever-present reality. Surface and groundwater in the hydrographic basins of northern Chile are scarce and farmers take care of what little they have. Therefore, considering all these aspects of Chile, it is possible to say that the historical Chilean practice of tailings management is outstanding in many aspects considering governance. The Chilean experience in tailings management offers many lessons for other countries facing the challenge of modern mining, given the current demand for safe tailings storage facilities.

Chile had disasters and tailings dam failures that impacted the environment and the community. However, after each failure, Chile’s legislators, regulators, consultants, and mining industry moved quickly and proactively to improve. This brought, as a consequence, changes and discarding of old construction methods and inappropriate operations. Chile learned from its mistakes and also from tailings management practices in other places, applied them, adapted them to its reality and improved them. In fact, it can be said that Chile is now a world leader in the application of the concepts of: (i) classification of tailings by hydrocyclones (separation of sand and slimes), (ii) thickened tailings, (iii) filtered tailings, (iv) construction of high tailings dams and (v) earthquake-resistant tailings storage facilities [8].

Copper mining is an important contributor to the economy of several developing regions in Chile. Many sites of mining interest in the country have abrupt geographies and potentially limited land area for tailings disposal. There is competition for land use between agriculture, livestock production, cities, and mining. While the limited land area for tailings disposal may drive consideration of the ocean as an alternative, it is important to recognize that local communities depend on the ocean as a major supplier of food and many livelihoods [9]. Environmental impact assessments of tailings disposal on land and at the bottom of the ocean are usually limited by budgets and time frames, resulting in a limited capacity to understand longer-term environmental risks to underground aquifers and marine ecosystems, including interactions between land and sea ecosystems (food chains) [10,11,12,13].

This article presents a review of the progress toward the development of a holistic tailings management strategy that will involve a more cross-disciplinary, whole systems approach to tailings governance. This strategy should be perceived as creating global leaders: decision makers who actively shape our future with both proven technical ability and creative, cost-effective, environmental rationality, and innovative management of the complex social, economic, environmental, and communication aspects of mining projects. This review also describes recent proposals for the best available technologies (BATs), case histories of Chilean TSF using conventional technology, thickened tailings, paste tailings, filtered tailings, water use reduction, tailings reprocessing to obtain rare earth elements (REEs), circular economy, submarine deep-sea tailings disposal, and ways to avoid failure in a seismic region.

## 2. History of Evolution of Tailings Management and Governance in Chile

### 2.1. Tailings Management in the Beginning 20th Century

The beginning of the 20th century (1901) coincided with the invention of the flotation metallurgical process as mentioned above, which allowed the separation of copper sulphide minerals from the components that form the original mineral rock. The emergence of this metallurgical process was key to the exploitation and development of low-grade mines and the beginning of the generation of mining tailings [5].

The history of tailings governance in Chile, considering a large production in metric tons per day, begins at the El Teniente Mine, where industrial copper exploitation began for the first time in 1905 by the American company Braden Copper Company. At the El Teniente mine, around 30,000 metric tons per day (mtpd) of tailings were generated in 1915 [14].

To prevent damage to agriculture that could be caused by the discharge of tailings into the rivers, the mining company El Teniente built two tailings storage facilities called Arenas and Marga with the tailings disposed of in the Coya riverbed (Figure 2). These tailings storage facilities were later washed away by successive floods in August 1913, November 1914, January 1915, and June 1960 [15]. The tailings stored in the tailings storage facilities after these floods flowed down the Coya River and reached the Cachapoal River, causing the tailings to flow into the irrigation canals of many farmers in the area [15].

The first record of a tailings storage facility collapse in Chile resulting from an earthquake was the failure of the Barahona Tailings Dam in 1928 (Figure 3). The tailings storage facility was located in Mina El Teniente, approximately 180 km from the epicenter in the city of Talca, with an earthquake measuring 8.2 on the Richter scale in central Chile. The failure of this tailings storage facility on 1 December 1928 was the first recorded case of dynamic liquefaction-induced failure of a tailings storage facility in Chile [16]. The tailings dam was destroyed as a result of this earthquake, and the resulting flow of liquefied tailings inundated downstream settlements and facilities, killing 54 people. The seismic event, known as the Talca Earthquake, also caused severe damage to houses, industrial facilities, and fatalities in the neighboring provinces of Talca, Curicó, Colchagua, and Cachapoal [17]. The tailings spill caused by the earthquake also caused damage to agriculture, destroying some fields of agricultural crops.

In February 1939, after a series of technical studies and site evaluation, the Cauquenes lagoon began to be used as a tailings storage facility. The tailings generated in the concentrator plant were transported to the tailings storage facility by means of a wooden channel, which was built with a 42 km route from the concentrator plant in the mountains to discharge into the Cauquenes’ tailings storage facility. The tailings in the Cauquenes sector were classified by separating their fine and coarse fraction using hydrocyclones: mechanical devices that use gravity to separate the coarse (sands) and fine particles of the tailings (slimes). The coarse particles were used to build the dam and the fine particles were stored in the reservoir area [15].

Most of the copper mines in Chile are located in the central and northern Andes Mountains. Ore extraction, crushing, milling, metallurgical processing, and tailings management operations normally take place around underground or open-pit mines. For example, the Potrerillos Copper Mine disposed of mine tailings near its processing plant when it began operations (1920–1938). However, the copper tailings were subsequently disposed of in coastal areas using a technique called offshore tailings disposal (disposal of riverine and marine tailings). This activity was carried out by the copper mines called Mina Potrerillos (1938–1958) and Mina El Salvador (1958–1990), located in the Third Region of Atacama Chile. The El Salvador Mine is located between 2400 and 2600 m above sea level (masl) and is 120 km east of the coastal city of Chañaral. According to historical records, for 52 years (1938–1990), untreated tailings (solids, water, and chemicals) from copper mines called Potrerillos and Salvador were directed to the Salado River bed (tailings disposal in the river). These tailings were subsequently deposited directly on the sandy beach of Bahía de Chañaral and Caleta Palitos, located on the Pacific Ocean coast (offshore tailings disposal). These sites mentioned in Chañaral Bay and Caleta Palitos received 300 and 125 million metric tons of copper tailings, respectively, currently being sites contaminated with high concentrations of heavy metals (As, Pb, Mo, Mn, among others) and chemical reagent residues (pine oil, xanthates, cyanide, flocculants, among others) (Figure 4) [18].

### 2.2. Advances and Failures in the Governance of Tailings in the Mid-20th Century

At the beginning of the 20th century, the main technological advances in Chilean tailings governance consisted of: (i) the hydraulic transport of tailings considering long distances (approximately 50 km) using pipes (steel) and channels (reinforced concrete and wood), and (ii) the construction of dams of the tailings storage facilities with the use of cycloned tailings sands. This served to prevent spills and the use of rivers for tailings transport. A 7.5 Richter magnitude earthquake occurred on 28 March 1965, causing a catastrophic failure of a major tailings dam at the El Cobre Tailings Dam, located at the El Cobre Mine (Figure 5). The failure of the El Cobre Tailings Dam caused more than 200 deaths and great material damage in industries and agriculture [17]. However, the construction of tailings dams up to that date consisted of the construction of dams through the construction method known as upstream direction. These tailings storage facilities were designed and built by experienced mining operators using the coarse fraction of tailings (cycloned tailings sand). The above is a dams’ construction method for tailings facilities that remained with little variation until 1965, when the tragedy of the El Cobre tailings storage facility occurred [17]. This tragedy made it clear that the technology and construction method of the time could no longer be used. For this reason, substantial changes were necessary to improve the safety of tailings storage facilities and allow the mining industry to continue in a responsible and safe manner.

It is important to mention that at that time, the science of soil mechanics applied to mining was still in an emerging phase and there were very few engineering specialists in soil mechanics in Chile. This event marked a milestone where the mining industry understood that the construction of tailings storage facilities required the experience of highly technically trained personnel, which exceeded the knowledge of only the miners. Due to this, it is possible to say that the participation of geotechnical experts in the design of tailings storage facilities began in the late 1960s. Since most of the geotechnical engineering knowledge at the time was gained from the design and construction of water dams, the first reaction applied to tailings storage facilities was to design dams that used borrowed materials from quarries and were consistent with existing water dam practice to contain mine tailings. Considering that time, it is important to remember that the El Cobre [17] and Fort Peck [22] tragedies discouraged the construction of tailings storage dams using cycloned tailings sands. Therefore, several embankment dams of the type used in water reservoirs were built during the 1970s. Examples include: (i) the Colihues tailings storage facility (a 83 m high embankment dam), (ii) the El Indio tailings storage facility (a 74 m high embankment dam), and (iii) the Los Leones tailings storage facility (a 160 m high water dam design) (Figure 6) [23]. However, despite this, the use of dams built with cycloned tailings sand continued during this period, with some modifications in their design and construction, mainly avoiding the use of the upstream construction method and applying the downstream construction method. Examples of cycloned tailings sand dams built during this period include El Cobre Tailings Dam No. 4 (El Soldado Mine), Cauquenes Tailings Dam (El Teniente Mine), and Tailings Dam No. 1 Pérez Caldera (Los Bronces Mine).

### 2.3. Mine Tailings Governance Improvements in the Late 20th Century

The 1980s for mine tailings governance was a period of technological advances in the study of cycloned tailings sand conditions as a construction material for tailings storage facility dams. The most striking and iconic design of this period is the Las Tórtolas tailings storage facility dam (Figure 7), a key work for the continuity of the Los Bronces Mine. This mine presented the following challenges: (i) transportation of ore through a 58 km pipeline from the mine in the mountain range to the concentrator plant located in the valley, (ii) a processing plant in the valley (far from the mine), and (iii) a 150 m high cycloned tailings sand dam located 40 km from Santiago, the capital of Chile. An engineering work of these characteristics, being a cycloned tailings sand dam of this height located in the vicinity of Santiago, materialized many studies on characterization, design, layout schemes and stability risk analysis of cycloned tailings sand [17]. It is also important to note that the design of the El Torito tailings storage facility (at the El Soldado Mine), which replaced the El Cobre Tailings Storage Facility No. 4, involved the use of cycloned tailings sands.

This time period during the 1980s marked the widespread adoption of cycloned tailings sand dam designs considering downstream and centerline construction methods. It was later found that the cycloned tailings sand dams successfully withstood in the 7.8 Richter magnitude earthquake of March 1985. Considering these experiences, it was found that this type of dam design maintains stable cycloned tailings sand saturation in the dam during high-intensity seismic events.

In the late 1980s in mine tailings governance, a new engineering design and construction trend emerged involving the comprehensive management of tailings and mine waste rock materials. Until then, the use of mine waste rock as construction material for tailings storage facilities was not considered and applied. During this period, the Romeral tailings storage facility (in the Romeral Iron Mine) was built next to an existing mine waste rock dump, and the dam was built with reject material from the metallurgical concentrator plant [26]. Another similar case began in 1989 with the Candelaria tailings storage facility (in the Candelaria Mine), where the location of the reservoir and the material for the dam were part of the design of the mine waste rock dump. During the second half of the 1990s, these experiences served as the basis for the design and construction of the Pampa Pabellón tailings storage facility at the Collahuasi Mine.

In March 1990, considering an Amparo Appeal requested by the community of Chañaral, the Supreme Court of Justice of Chile ruled against Codelco’s El Salvador Mine, and with this, the prohibition of the disposal of untreated mine tailings was enacted. Codelco’s El Salvador Mine was forced to build a tailings storage facility in the Atacama Desert called Pampa Austral. This was done at Pampa Austral, a deserted area near the town of Diego de Almagro. The so-called “clear tailings water” originating at the Pampa Austral TSF was legally channeled back to the bed of the Salado River and disposed of at Caleta Palitos starting in 1991. To date, no environmental rehabilitation was carried out, and beach disposal of tailings generates substantial dust emissions under windy conditions, which transport heavy metals to the town of Chañaral, posing a significant health risk to the population [27,28].

In the mid-1990s, following the successful performance of cycloned tailings sand dams during the March 1985 earthquake, the design of cycloned tailings sand dams was challenged to reach greater heights. The first case was the Quillayes cycloned tailings sand dam of the Los Pelambres Mine, located in the Andes Mountains, with a height of 175 m. This design incorporated all the innovations introduced to date and the results of the first studies on the effect of high confining pressure on the behavior of cycloned tailings sands, including the first three-dimensional (3D) dynamic analysis of a cycloned tailings sand dam [29,30]. A few years later, considering engineering studies, the height of this dam was increased to 198 m to prolong the useful life of the tailings storage facility, given the increase in ore production at the metallurgical processing plant. As a consequence of filling the Quillayes tailings storage facility with mining tailings for years, it reached its maximum storage capacity, for which a study had to be carried out to select a new site to store mining tailings. This is how the Mauro tailings storage facility was designed to allow the continuous operation of the Los Pelambres Mine (Figure 8). This tailings storage facility, in operation since 2008, is located 60 km from the metallurgical concentrator plant, with a cycloned tailings sand dam with a final maximum height of 237 m, the first to exceed 200 m in a region of high seismicity [31]. Currently, there are designs for tailings storage facilities located in Chile that are close to 300 m in ultimate height, and technical engineering tools are available to assess the physical stability of dams of that height. Great progress was made in increasing the geotechnical knowledge base on the use of cycloned tailings sands under high confining pressures in high dams.

At the end of the 1990s, the use of thickened and paste tailings began in Chile’s tailings governance. Tailings thickening was already applied in some mining operations in Chile as of the 1980s, considering low levels of thickening and without the use of flocculants, only gravitational processes. For example, the thickening level was less than 40% of the solids content by weight (Cw), modeling the hydraulic and rheological behavior of the tailings slurry as Newtonian, that is, without yield stress. The Paste and Thickened Conference in Chile in 2002 gave a tremendous boost to this new technology, considering the experiences applied in Canada, Australia, and South Africa, and it began to be used as a technically feasible and safe option. Initially, this new technology was reluctantly accepted by the low-production operations (Las Cenizas, Planta Delta, Alhue, and Planta Demo Collahuasi) and later its application was generalized, as in the rest of the world. Important advances in thickening equipment (conventional, high-rate, high-density, and deep cone thickeners) and the use of flocculants promoted the implementation of tailings thickening at high densities, reaching solid contents above 50% (Cw), modeling the hydraulic and rheological behavior of the tailings slurry as non Newtonian with yield stress. However, large mining operations were reluctant to apply this technology. Minera Esperanza (currently Minera Centinela), which began operations in 2011, took a big step in this direction by adopting thickening technology for tailings management, with a target solid concentration of 69% (Cw) and a tailings production rate of 90,000 mtpd. Recently, the Sierra Gorda Mine (located a short distance from the Centinela Mine) started applying it at a lower solid concentration of 63% (Cw) and a tailings production rate of 110,000 mtpd. Codelco carried out studies to implement thickened tailings, and thus disposed its tailings with greater density in the Talabre tailings storage facility, which will be implemented in the coming years reaching a tailings production rate of 400,000 mtpd with a solid concentration of 67% (Cw). One of the main problems associated with high concentration thickening operations is the estimation of the slope of the tailings beach, which is generally overestimated. Usually, according to practical experience in some mining operations, thickened tailings deposition slopes are of the order of 2.0% to 4.0%.

## 3. From the Lessons Learned to the Country’s Regulation

The tailings management development in Chile is closely linked to important seismic events from which lessons were learned over time. A timeline of the history of mining tailings management and governance in Chile is presented in Figure 9, highlighting some main aspects related to earthquakes, failures of tailings storage facilities, construction heights of dams of some tailings storage facilities, and the evolution of tailings management technology.

### 3.1. Improvements in Tailings Management and Regulation: Supreme Decree No. 86 of 1970

The March 1965 earthquake prompted widespread discussion in Chile and public pressure for stricter control over tailings dams. The Chilean authorities issued Decree No. 86 in 1970 to regulate the design, construction, and operation of tailings dams. This regulation provided definitions, criteria, and specifications for the design, construction, and operation of tailings dams, including periodic monitoring activities. One special clause of this decree was the prohibition of the tailings dam upstream construction method [17].

The study of the accident that occurred at the El Cobre tailings storage facility established the need to modify the construction method of tailings storage facilities with cycloned tailings sand dams. This resulted in the creation of a regulation to be applied in mining, which was codified No. 86 (1970), where, for the first time in Chile, there were definitions, criteria, restrictions, limitations, and requirements for the design, construction, and operation of tailings storage facilities. This regulation explicitly prohibited the use of the upstream construction method in seismic areas in Chile because it was shown that dynamic liquefaction occurred in the tailings, which was an important step in the practice of tailings management and is exclusive to Chile. It should be noted that this decree only addressed those tailings storage facilities whose dams were built with cycloned tailings sand, excluding dams built with borrowed materials (earth embankments or rockfill dams). Tailings storage facilities built with cycloned tailings sand dams were implicitly recognized as posing the greatest risk, although they were possibly the most common dam types at the time. The new regulation incorporated key geotechnical concepts regarding: (i) the resistance of the materials (cohesion and internal friction angle parameters) and (ii) the effect of water on the materials (moisture content and permeability). In addition, it initiated a new generation of tailings storage facilities with cycloned tailings sand dams where the sand had to meet minimum placement density and permeability specifications to be used as a construction material, and the dam had to have a solid base drainage. Examples of dams built during this period to meet these requirements include: the Pérez Caldera Tailings Dam No. 2 (at the Los Bronces Mine), the Piuquenes Tailings Dam (at the Andina Mine), and the expansion El Cobre Tailings Dam No. 4 (at the El Soldado Mine).

In this evolutionary engineering process, there is a particular case: The Talabre tailings storage facility in Chuquicamata, where a site was selected to store the tailings, which was an old Salar (Figure 10). The design of this dam was completed in the late 1970s and construction began in 1983. The dam was built with cycloned tailings sand and with the following details (different from the current design types): (i) an intermediate sloped drainage within the dam; (ii) a downstream slope of 2.5:1.0 (H:V); and (iii) a method of constructing a paddock in the downstream direction with mechanical transport and placement of sand from cycloned tailings (rather than the commonly used hydraulic method) [23].

### 3.2. First TSF Seismic Performance with Supreme Decree No. 86 of 1970 Regulation—3 March 1985

Three other cases of seismic effects occurred at three tailings dams in central Chile after the earthquake of 3 March 1985, a 7.8 Richter magnitude earthquake with an epicenter approximately 120 km away in the city of San Antonio. In the first case, the Veta de Agua No. 1 tailings dam failed, and the stored fines tailings moved along the El Sauce Creek for approximately 5 km [17]. In the second case, the Cerro Negro No. 4 tailings dam failed, losing 25% of the dam structure [17]. In the third case, the El Cobre No. 4 tailings dam developed cracking along the dam crest [17]. It is noteworthy that in contrast to previous earthquakes, the 1985 earthquake caused damage only to small-scale mining operations, and no deaths were reported. However, the authorities decided to increase the construction controls on tailings dams, emphasizing the fines content, compaction, and drainage of cycloned tailings sand used for dam construction.

### 3.3. Reassessment of Current Chilean Tailings Storage Facilities Regulation: Supreme Decree No. 248 of 2006

In Chile, the National Services of Geology and Mining (SERNAGEOMIN) provided the legal text for Supreme Decree No. 248 from 2006, which regulates the approval of the design, construction, operation, and closure of tailings storage facilities [35]. This regulation replaced Supreme Decree No. 86, which was established after the earthquake in 1965 that caused the failure of several tailings dams constructed using the upstream method (currently prohibited due to seismic conditions).

The current regulations regulate all aspects of engineering and construction of mining tailings deposits in Chile. The emphasis of the regulations is to guarantee the physical stability of tailings deposits considering the seismic condition of Chile. Geotechnical aspects, such as dam construction material properties, dam stability analysis, and controlled management of stored tailings are a priority for the regulations [35].

However, the risk associated with TSFs is very large, considering that a potential dam failure could easily damage the environment and people with pollution or other adverse impacts [36]. Accidents or failures at these facilities are always associated with social, environmental, or public safety issues, some more catastrophic than others [37]. This contributed to a reduction in the rate of TSF failures in recent decades. In most countries around the world where responsible mining is practiced on a day-to-day basis, regulatory frameworks became more stringent, requiring a higher level of responsibility, with the intent of minimizing risks posed to society and the environment.

From 2014 to date, there were major disasters worldwide as a result of tailings deposit failures. Some examples are Mount Polley Canada (2014), Fundao Samarco Brazil (2015), Corrego do Feijao Brumandinho Brazil (2019), and Jagersfontain South Africa (2022), which caused the death of hundreds of people and irreparable environmental damage [16]. As a consequence of these disasters, the international community, institutions and global groups such as the International Council on Mining and Metals (ICMM), UN environment program, and the Principles for Responsible Investments (PRI), developed a “Global Tailings Management Standard for the Mining Industry”, launched in August 2020, with the aim of regulating the operation throughout the entire life cycle of tailings deposits, including closure and post-closure (perpetuity), considering zero harm to people and the environment, and zero tolerance for human fatalities [2]. ICMM advocates that the application of appropriate design and management standards and good practices allow tailings storage facilities to be safe. ICMM members are committed to preventing catastrophic failures of tailings deposits, with continuous improvement in the design, construction, and operation stages of these facilities. This organization urges mining companies to improve their management by adopting the Global Industry Standard on Tailings Management, taking advantage of technological innovation and continuous improvement [2].

In this context, the current Chilean regulatory framework needs reassessment to consider the best available practices for tailings storage and causes of tailings dam failures around the world, with the objective of establishing guidelines, restrictions, and incentives to implement environmentally friendly solutions [38]. Topics that need reassessment include the following:Definitions of environmental criteria for TSF site selection.Incentives and restrictions for the use of dewatering, thickening, paste, and filtered tailings technologies.Evaluation of the lining of the storage area of the tailings deposit to prevent leaks into the soil and the environment.Geochemical characterization, prediction, and mitigation specifications, tests, and criteria for managing acid rock drainage (ARD).TSF dust mitigation parameters during operation, closure, and post closure (perpetuity).Integrated assessment criteria between SERNAGEOMIN, the General Directorate of Waters (DGA), the Ministry of Health (MINSAL), and the Environmental Assessment Service (SEA).Submarine deep-sea tailings disposal regulatory framework (definitions, restrictions, limits, or prohibition of method).Regulation of reprocessing and circular economy of old TSFs (definitions, restrictions, metallurgical technologies, and limits).Baselines for risk analyses and territorial strategic planning for operating and abandoned TSFs.

The focus of this reassessment should be on tailings dam safety legislation, regulatory frameworks, and specific guidelines associated with the stewardship of TSFs [38].

### 3.4. Second TSF Seismic Performance with Supreme Decree No. 248 of 2006 Regulation—27 February 2010

No other large earthquake events occurred until the 8.8 Richter magnitude earthquake at midnight on 27 February 2010, the most powerful earthquake recorded in the area and the second most powerful in the country’s history after the 1960, 9.5 Richter magnitude earthquake in Valdivia city. Based on reports presented by national and international experts, the following three main observations and conclusions were drawn regarding the impact of the 2010 earthquake on tailings deposits: (i) Compared to the Barahona Tailings Dam failure in 1928, tailings dam failures in the 1965 La Ligua earthquake, and two failures in the 1985 San Antonio earthquake; the performance of large tailings dams during the operation phase seems to show some improvement [39]. (ii) Eyewitness reports indicate that the tailings in the basins of the deposits of some of the facilities (Ovejería, Las Tortolas, Carén, Los Leones) “presented local liquefaction in tailings beaches during this event, with waves of tailings clearly evident and contained by the dam crest freeboard” [17]. (iii) Some small abandoned tailings dams located in central Chile suffered damage due to liquefaction, three tailings dams constructed using the upstream method (e.g., Chancón, Bellavista Dike No. 1, and Veta del Agua Dike No. 5) experienced varying levels of seismically induced failure, and the Las Palmas Tailings Dam collapsed, killing four persons [17,19]. Overall, the seismic performance of large tailings storage facilities in operation during the 8.8 Richter earthquake in central Chile in 2010 proved to be satisfactory, without the breaking of dams or significant changes in the form of the tailings impoundments.

## 4. Copper and Mine Tailings Production in Chile

### 4.1. Copper Production in Chile and Worldwide

Chile has 28% of the world’s copper reserves, more than twice the reserves of neighboring Peru, the world’s second leading copper producer. Chile has reserves of 190,000 million metric tons of copper, 26% more than previously thought (150,000 million tons), which will allow copper extraction for the next 100 years at the current extraction rate, according to the United States Geological Survey [7]. The constant increase in copper prices in recent decades made Chile’s economy one of the strongest and most robust in Latin America, since the red metal represents more than 40% of the country’s exports and its main source of income. Chile currently produces more than a third of the copper produced worldwide [7].

Figure 11 shows the rate of copper production in millions of metric tons per year from 2011 to 2018 for the leading copper-producing countries in the world today: Chile, Peru, China, the United States, and Australia.

As Figure 11 shows, Chile is by far the world’s leading copper producer, with more than 5 million metric tons of copper currently produced per year, followed by China, Peru, the United States, and Australia, with annual production levels between 1.0 and 1.5 million metric tons per year. The explosive growth of copper production in Chile began at 1990 and is attributable to the consolidation of the government mining company Codelco, the entry of foreign private mining companies into Chile, free-market economic policies, high demand for copper worldwide, and the high prices of copper commodities.

### 4.2. Historical Growth of the Chilean Economy and Relantionship with Copper Mining

Chile has a high-income market economy according to rankings produced by the World Bank [40]. The country is considered one of South America’s most prosperous nations, leading the region in competitiveness, income per capita, globalization, economic freedom, and a low level of perceived corruption. Chile is among the most industrialized countries in Latin America. Some of its key industries are copper mining, product manufacturing (food processing, chemicals, and wood), and agriculture (fishing, viticulture, and fruit) [40].

In 2006, Chile became the country with the highest nominal gross domestic product (GDP) per capita in Latin America. Gross domestic product (GDP) is a monetary measure of the market value of all the goods and services produced during a specific time period by a country. GDP is often used as a metric for international comparisons and a broad measure of economic progress. It is described as the “world’s most powerful statistical indicator of national development and progress” [41]. The industrial sector in Chile contributes 31.4% of the GDP and employs 22.2% of the working population. The mining sector is one of the pillars of the Chilean economy, mainly because of the country’s enormous copper reserves and production of more than a third of the world’s copper annually. Figure 12 shows the historical evolution of Chile’s GDP per capita from 1810 to 2018.

Figure 12 shows the explosive growth of the Chilean economy in the last 35 years, which resulted mainly from the growth of copper mining following the consolidation of the government mining company Codelco and the entry of private mining companies with foreign capital into Chile. As discussed later in this article, the growth of the economy and the growth in copper production are directly related to the growth in mining tailings production.

### 4.3. Historic Mine Tailings Production in Chile

To understand and measure the evolution of tailings governance in Chile, historical mine tailings production per year was estimated. Information from the National Services of Geology and Mining (SERNAGEOMIN) registry on mining tailings in Chile [42] was complemented with information from the Consejo Minero tailings platform [24] to identify, characterize, and locate the distribution of large-scale tailings storage facilities between Region I (Tarapaca) and Region VI (O’Higgins). Figure 13 shows the total amount of mining tailings produced per year between 1905 and 2022 corresponding to large-scale tailings storage facilities in Chile.

Chile currently produces 800 million metric tons of mining tailings per year, making it one of the largest producers of this mining waste worldwide. It is important to note that the metallurgical processing concentrator plants that generate tailings operate 24 h a day, continuously generating tailings, even at night. Chile’s current annual mine tailings production level, shown in Figure 13, corresponds to the production of approximately 2,192,000 metric tons of mining tailings per day.

Chile is divided into three zones: north, centre, and south. Additionally, Chile is divided into 16 regions, as shown in Figure 14. This image displays the spatial distribution and location of large mining projects and the presence of mining tailings deposits in the country.

Most of the copper mining projects in Chile are located in the north, mainly in the Tarapaca I, Antofagasta II, Atacama III, and Coquimbo IV regions. Another part of the large mining projects is located in the central zone of the country, in the regions of Valparaiso V, Metropolitan RM, and O’Higgins VI. Looking at the map it is possible to understand the limitation of space available to carry out activities and land use planning related to: mining, agriculture, livestock, industry, ecosystem conservation, and cities.

Based on information from SERNAGEOMIN [42] and the Consejo Minero [24], Table 1 shows a summary of the distribution of mining tailings by region, indicating: (i) the number of tailings storage facilities, (ii) the number of tailings storage facilities with government (Coldelco) and foreign private investment, (iii) the quantity of total tailings deposited (millions of metric tons) corresponding to government (Coldelco) and foreign private investment, (iv) the total volume of tailings deposited (millions of m^3^), and (v) the area covered by mining tailings (Ha).

Table 1 shows that the total amount of mining tailings corresponding to large-scale tailings storage facilities in Chile generated between 1905 and 2022 was equivalent to 13,021 million metric tons or its volume equivalent of 8494 million m^3^, corresponding to a total surface area of 26,876 Ha.

Based on information from SERNAGEOMIN [42] and the Consejo Minero [24], Figure 15 shows the distribution of the total amount of tailings (in millions of metric tons) stored in large tailings storage facilities in Chile by region from 1905 to 2022. The figure shows the total tonnage of tailings stored in large tailings storage facilities operated by the government (Codelco) and private companies in each of the seven regions.

As Figure 15 shows, the region with the largest amount of stored tailings is Region II (Antofagasta), followed by Region VI (O’Higgins). The government mining company Codelco has a total of 6654 million metric tons of tailings stored, exceeding the total of 6328 million metric tons of tailings that the foreign private mining companies have stored. The amount of tailings stored in Region II (Antofagasta) by the government mining company Codelco corresponds to the tailings deposited in the Talabre tailings storage facility in the vicinity of the city of Calama from concentrator plants of: Chuquicamata, Ministro Hales, and soon Radomiro Tomic. The amount of tailings stored in this same region by foreign private mining companies corresponds to the tailings deposited in: (i) the Hamburgo, and Laguna Seca tailings storage facilities operated by the company Minera Escondida, (ii) Esperanza tailings storage facility operated by Minera Centinela, (iii) Catabela tailings storage facility operated by Minera Sierra Gorda, (iv) Spence tailings storage facility operated by Minera Spence and (v) Fine and Filtered tailings storage facilities operated by Minera Mantos Blancos. In Region VI (O’Higgins), the tailings storage facilities that contribute to the storage of tailings are the Carén, Colihues, Cauquenes, Barahona 0, Barahona 1, and Barahona 2 impoundments located in the vicinity of the city of Rancagua.

The amount of tailings stored in Region III (Atacama) by the government mining company Codelco corresponds to the tailings deposited in the Potrerillos Plant, Chañaral Bay, Caleta Palitos Bay, and Pampa Austral tailings storage facilities. The amount of tailings stored in this same region by foreign private mining companies corresponds to the tailings deposited in: (i) the Candelaria and Los Diques tailings storage facilities operated by the company Minera Candelaria and in (ii) the La Brea and Sand Stack tailings storage facilities operated by the company Minera Caserones.

The amount of tailings stored in Region RM (Region Metropolitana) by the government mining company Codelco corresponds to the tailings deposited in the Ovejería tailings storage facility in the vicinity of the city of Santiago. The amount of tailings stored in this same region by foreign private mining companies corresponds to the tailings deposited in the Perez Caldera No. 1, Perez Caldera No. 2 and Las Tortolas tailings storage facilities operated by the company Minera Los Bronces. In Region IV (Coquimbo), the tailings storage facilities that contribute to the storage of tailings are: (i) Los Quillayes and El Mauro tailings storage facilities operated by Minera Los Pelambres, and (ii) Andacollo tailings storage facility operated by Minera Carmen de Andacollo.

The amount of tailings stored in Region I (Tarapaca) by foreign private mining companies corresponds to the tailings deposited in: (i) the Pampa Pabellon tailings storage facility operated by the company Minera Doña Ines de Collahuasi and (ii) the Quebrada Blanca tailings storage facility operated by company Minera Quebrada Blanca Phase II. The amount of tailings stored in Region V (Valparaíso) by the government mining company Codelco corresponds to the tailings deposited in the Los Leones and Piuquenes tailings storage facilities in the vicinity of the city of Los Andes. The amount of tailings stored in this same region by foreign private mining companies corresponds to the tailings deposited in the El Cobre No. 4, and El Torito tailings storage facilities operated by the company Minera El Soldado.

Finally for this analysis, Figure 16 and Table 2 show the 10 large-scale tailings deposits for each region of Chile, considering the information available to the year 2019. In Figure 16 it is possible to appreciate the size of each tailings deposit and its location in the country, while in Table 2, it is possible to observe the current capacity of stored tailings and the capacity authorized by SERNAGEOMIN.

## 5. Current Chilean Tailings Governance Practices: Use of the Best Available Technologies (BATs)

The term “best available technologies” (BATs) refers to “the most effective and advanced stage in the development of activities and their methods of operation which indicate the practical suitability of particular techniques for providing, in principle, the basis for emission limit values designed to prevent and, where that is not practicable, generally to reduce emissions and the impact on the environment as a whole” [33].

According to this definition, BATs applied to the management of TSFs considers the following aspects: (i) reliable performance of technologies, (ii) controlled storage of tailings (considering site-specific conditions), and (iii) efficient water management (including the control of water losses due to evaporation, retention in interstitial voids of tailings, and infiltration) [45]. If these key issues are successfully addressed, a reduction in water make-up requirements and a decrease in negative environmental impacts can be achieved, signifying the implementation of environmentally friendly tailings management [33].

Currently, many major copper mining projects located in extremely dry areas, such as northern Chile in the Atacama Desert, process copper sulphide ores at high production rates: in some cases, more than 100,000 metric tons per day (mtpd), as are the cases of: (i) Escondida Mine, (ii) Chuquicamata–Ministro Hales–Radomiro Tomic Mines, (iii) Centinela Mine, (iv) Sierra Gorda Mine, and (v) Spence Mine. The result is the generation of large amounts of mine tailings that are commonly handled and transported hydraulically in pipelines to tailings storage facilities (TSFs) using either freshwater or seawater. Originally, mining companies used water from two main sources: (i) underground water and (ii) river water. With the passage of time and the effects of climate change, these resources were depleted, which forced mining companies to seek water from the sea and apply mining tailings dewatering technologies.

Considering the extremely dry climate, the scarcity of water, the demands of the community for access to this important resource, as well as the environmental limitations in these desert areas of Chile, the efficient use of water in mining is imperative. Therefore, water supply is recognized as one of the restrictions for the development of new mining projects and the expansion of existing ones in these areas. The mining companies are developing strategies for new water supply alternatives, such as water recovery from tailings and seawater (desalinated or not), to address this growing scarcity [8].

The transport and storage of tailings requires governance with an environmental focus. This waste is normally handled and transported hydraulically using water almost 365 days a year and 24 h a day (there are only a few days a year when the concentrator plant is stopped for maintenance of the grinding and flotation equipment). This alternative is cheaper than transportation of bulk solids by conveyor belts, trains, or trucks. For hydraulic transport to be a sustainable and environmentally friendly practice, most of the water used for tailings transport must be recovered for reuse in the metallurgical flotation process [8].

Mining production rates at copper sulphide deposits are increasing significantly due to declining copper grades at existing mines. As copper ore grades decline, more mine waste rock is generated and more ore with valuable metals needs to be processed to produce the same amount of copper. The use of water in the metallurgical mining process of grinding and flotation is proportional to the amount of ore that is processed, so it follows that more water is needed to produce the same amount of copper as grades decrease. The exploitation of large deposits of copper ore with decreasing grades led to the need to use large and efficient equipment for grinding and processing the ore in flotation cells to obtain higher production rates, which in turn implies a greater demand for minerals and water for metallurgical processes [8].

The application of tailings dewatering technologies to increase water recovery from tailings is an appropriate step to reduce water losses in tailings storage facilities caused by evaporation, infiltration, and retention in pore voids. The use of tailings dewatering technologies promoted the improvement of the physical stability of tailings deposits, because the tailings material has a lower water content, reaching higher levels of consolidation faster. Figure 17 shows new designs and technologies applied in Chile that consider the tailings continuum principle to achieve environmentally friendly tailings management with a focus on efficient water use [46].

Considering the best available technologies (BATs) for tailings management incorporating tailings dewatering techniques, it is possible to mention four main categories: (i) conventional tailings, (ii) thickened tailings, (iii) paste tailings, and (iv) filtered tailings [48].

Conventional copper tailings typically range 25–40% solid weight concentrations (Cw), thickened copper tailings 40–65% solid weight concentrations (Cw), paste copper tailings 65–80% solid weight concentrations (Cw), and filtered copper tailings over 80% solid weight concentrations (Cw) (solid concentrations may vary with particle size and shape, clay content, mineralogy, electrostatic forces, and flocculent dosing).

Conventional, thickened, paste and filtered tailings refers to a continuum of tailings with high solid concentrations and higher yield stress, due to the greater level of fluid removal from tailings before disposal.

### 5.1. Conventional Tailings Disposal

In current large-scale Chilean mining operations in dry areas, most typical tailings disposal schemes consist of conventional disposal or thickening prior to disposal of solid weight concentrations (Cw) of 25–40% [8]. Conventional TSFs have dams built of cycloned tailings sands (coarse fraction of tailings obtained using hydrocyclones) or have slightly thickened tailings deposits with dams built of borrow material. Conventional tailing dams may have water recoveries as high as a 55–70% range in very well-operated TSFs, which means they have appropriate tailings distributions, good control of the pond (volume and location), and adequate seepage recovery [8,49]. In conventional dams, water decanting at the settling pond is recovered by floating pumps or decant towers, and dam seepage is collected by a drainage system and cutoff trench systems. Some mining operations using this technology are: Los Pelambres, Los Bronces, Salvador, El Soldado, and Andina (Figure 18 and Figure 19) [8,50].

### 5.2. Thickened Tailings Disposal

Thickened tailings disposal (TTD) technology revolutionized the mining industry and is an interesting alternative to conventional tailings. In a TTD tailings storage facility, the properties of the tailings and their location are “designed” to fit the topography of the disposal area. In conventional disposal, tailings are segregated as they flow and settle in a reservoir, whereas in TTD technology, a sloped surface is obtained without particle segregation [8]. Solids content by weight in the order of 40% to 65% can be achieved due to the evolution of tailings thickening equipment. In this way, TTD achieves high water recovery (70% of recovered tailings water content) and self-supporting tailings storage facilities with sloping beach surfaces, requiring small dams (Figure 20 and Figure 21). Some mining operations that use this technology are: Centinela, Sierra Gorda, Spence, Carmen de Andacollo, and Chuquicamata–Ministro Hales–Radomiro Tomic (this last case is in the feasibility phase of implementation in Talabre tailings storage facility) [8,53].

### 5.3. Paste Tailings Disposal

Paste tailings technology was applied on a small production scale due to the limited manufacturing capacity of thickening equipment called deep cone thickeners. This method makes it possible to satisfy a recovery of water from the tailings of the order of 80%. Paste tailings are dense, viscous mixtures of tailings and water that, unlike slurries, do not segregate when not in transit. Advantages of using paste tailings technology include the ability to: (i) recover water and process reagents, (ii) maximize tailings density, (iii) minimize tailings storage facility footprints, (iv) make tailings suitable for backfilling underground mines, (v) reduce the potential for acid drainage (by removing water available for leaching, lowering permeability and oxygen diffusion), (vi) minimize (or eliminate) the risks of failure, and (vii) the emission of dust is negligible because the surface of the deposited tailings remains as a solid hard crust due to the bonding of the tailings particles by the action of the flocculant [54]. However, in some cases, due to the high viscosity and non-Newtonian rheological behavior of paste tailings, there are difficulties in tailings transportation that require the use of positive displacement pumping, which generates high capital costs. The main advantage of this method is that large dams are not required; only small dams are needed and the stored tailings material forms a solid crust with steep deposition slopes of the order of 2.0% to 4.0% (Figure 22 and Figure 23). Some mining operations that use this technology are: El Toqui, Enami Delta Plant, Las Cenizas, Demo Plant Collahuasi, and Alhue [8,53].

### 5.4. Filtered Tailings Disposal

In the past 20 years, many mining projects around the world applied a new tailings disposal technology called filtered dry-stacked tailings. This technique allows the tailings to be drained such that a solid material is obtained, also called unsaturated cake, which allows the material to be stored without the presence of water and without the need to handle large tailings ponds. The application of this technology was achieved due to the advancement of the development of vacuum filters and pressure filters. The main advantages of this technology are the following: (i) increase in the water recovery from tailings (90%), (ii) reduction in the TSF footprint, and (iii) reduction in the risk of physical instability, being the TSF structures are self-supporting under compaction [47]. The improvements in filtering technologies (pressure and vacuum filtering) in recent years allowed increased operational reliability and the development of large-capacity filters, reaching 50,000 metric tons per day (mtpd) of filtered tailings in some projects (Figure 24 and Figure 25). Some mining operations using this technology are: El Peñon, Mantos Blancos, Huasco, El Gato, Salares Norte, Potrerillos, Tambo de Oro, Tambillos, El Indio, and La Coipa [47].

Table 3 summarizes the main characteristics of the tailings storage facilities of the large mining operations in Chile regarding the type of technology used, tailings production rate, type of dam construction material, and dam height projected.

As Table 3 shows, some mining operations have tailings production of more than 100,000 mtpd, led by the Escondida Mine, which is the largest copper mining operation in the world, with an approximate production of 370,000 mtpd of mining tailings. The tailings storage facilities for operations using conventional tailings technology have dam heights of more than 100 m in some cases, which is not surprising, considering the high filling rates and topographic conditions of the valleys in which these tailings storage facilities are located. Some cases stand out, such as El Mauro and Quebrada Blanca Phase II, with dam heights of more than 237 and 310 m, respectively. These are the highest tailings storage facilities dams in the world. The Caserones Sand Stack case does not correspond to a dam as such but rather to a stockpile of cycloned tailings sand material compacted on the slope of a ravine, projecting 500 m high from the base of the ravine to the crest of the stockpile. Finally, it is possible to mention that the tailings storage facility that will receive the largest amount of tailings produced of 400,000 mtpd will be the Talabre TTD TSF project, considering the thickened tailings disposal technology. Tailings production corresponds to the contribution of three concentrator plants: (i) Chuquicamata, (ii) Ministro Hales, and (iii) Radomiro Tomic. The implementation of this project will mark a worldwide milestone on the implementation of thickened tailings disposal (TTD) technology with management of mining tailings on this scale.

## 6. Evolution of Tailings Governance Paradigm in Chile: Road to a Holistic Approach

### 6.1. Only Physical TSF Stability Focus: “The Old School in Tailings Governance”

The reductionist paradigm appeared early in modern scientific development and is expressed when applying the Cartesian method. The ecosystem paradigm emerged in the 20th century as a result of the development of the general systems theory and the concept of ecosystems. The application of the reductionist paradigm can explain the presence of an economic view of natural resources in mining and mine waste management, particularly water, minerals, and tailings, among others. The application of the ecosystem paradigm helps explain a holistic view of these natural resources [56].

The main limitation of the reductionist approach applied in mining is its inability to explain complex and open systems, such as tailings storage facilities in spatial and time scenarios, which are mine wastes that interact with ecosystems. It is not possible to explain the behavior of a system simply by identifying its components, particularly when the components are nonlinearly related, such as tailings storage facilities [56]. During the 1960s, a strong engineering presence (mainly geotechnical, civil, hydraulic, and mining) was established in the design, construction, and operation of tailings deposits in Chile. The main concerns were the physical stability of tailings dams and the execution of designs using an efficient cost–benefit approach, without consideration of important environmental and social issues.

### 6.2. Physical–Hydrological–Geochemical TSF Stability Approach: “The New School, beyond Only Engineering in Tailings Governance”

Environmental issues are becoming increasingly significant in industrialized countries, mainly in Europe and the United States, together with various historical events that led to the reconsideration of the production dynamics and technological expansion developed in the framework of the modernization processes. An important milestone in this process was the Stockholm Conference, held by the United Nations in 1972, to discuss the environmental reality at that time and establish common principles for the preservation and improvement of the human environment, considering the natural and artificial aspects of its composition. This contributed to a global awareness phenomenon emphasizing the need to reduce the impacts on ecosystems and communities being caused as a result of the overexploitation of natural resources, such as minerals by mining [57].

Since the 1980s, the way of dealing with issues using a reductionist approach led to specialization in areas of research and theoretical elaboration from a disciplinary perspective. An effort was made to describe the properties of nature by identifying its components and simplifying their study to facilitate the interpretation of the results [56]. These advances were applied in the stages of design, construction, operation, closure, and reinstatement of mining projects. Some of the specialization areas that stood out the most for their socioenvironmental contribution to the development of mining projects over the last 20 years include the following disciplines:Environmental engineering (assessment of environmental impacts and land remediation).Geochemistry (the corresponding to acid rock drainage (ARD) prevention and mitigation).Hydrology (runoff water quantity and quality control, water cycle and climate change).Hydrogeology (underground water quantity and quality control).Ecology (care and preservation of biota and interactions with anthropogenic activities).Sociology (interactions with communities and understanding of human livelihoods).Geography (development and changes in society on different spatial and temporal scenarios).Economy (circular economy and capital/operational/closure cost estimate).

This is how, little by little, new approaches to addressing the most complex issues and assembling professional teams are generated in tailings management. This knowledge paradigm, called holism, promotes an interdisciplinary approach to applying the spatial–temporal scenarios to analysis issues, overlapping with a socioenvironmental approach.

### 6.3. Interdisciplinary, Spatial–Temporal Scenarios and Socioenvironmental Approach: “Road to Holistic Tailings Governance”

Socioenvironmental knowledge allows us to understand that environmental issues exist on multiple levels and involve interdependent relationships between society and nature. They must be comprehensively analyzed from the perspectives of different disciplines rather than observed through “filters” derived from professional or life experiences. Therefore, this interpretation perceives that tailings management needs a comprehensive vision. Currently in Chile, tailings management is not part of long-term spatial–temporal scenario planning within the country.

The situation in Chile, with respect to the social and economic importance of the mining industry and the environmental conflict it generates, is a consequence of the geopolitical and economic model of the country, which favors centralism and extractivism. There is a conflict regarding the use of the territory in central Chile, with society, industry, and government all needing space to develop their activities (land use planning). For this reason, the mining industry knows that no more space in central Chile is available to store mine tailings (on land).

Thus, environmental issues associated with tailings management heightened the need for holistic and systemic approaches to examine the interrelations between the different processes that influence and characterize this kind of issue. This demand stimulated the development of theories to find common homologies of different logics, articulating knowledge in interdisciplinary research methods for complex socioenvironmental analysis.

The management and governance of mining tailings must consider the entire metallurgical mining process as a complex, dynamic, and variable system in time and space, which means: (i) understanding the geology and mineralogy of the mining deposit, (ii) analyzing the geochemistry of the processed minerals, (iii) knowing the applied crushing and grinding processes, (iv) knowing about the chemical reagents applied in the flotation and thickening process, (v) knowing the quantity and quality of the water used in the metallurgical process (fresh water, underground water or sea water), (vi) understanding the climatic and seismic conditions of the mining area, (vii) knowing the natural ecosystems in the area of influence of the mining project, and (viii) having continuous dialogue with neighboring communities.

Implementing the ecosystem paradigm was suggested as a holistic approach to tailings governance that incorporates consideration of the relationship between the environment and human society. These approaches are essential to avoiding the application of only engineering criteria by incorporating an ecosystem-based approach, including the following principles: (i) inclusive participation of the community in water and tailings management issues, both in time and space; and (ii) the right to live in a pollution-free environment as a fundamental human right.

Although progress was made in recent years in developing a holistic approach to tailings governance, progress in regulations, applied technologies, and measures to improve tailings management are acknowledged and appreciated, but these advances are small, and it will probably take some years to address environmental issues with a holistic approach and a long-term vision.

There is thus potential for an integrated holistic approach to incorporate a greater level of sustainability into the tailings storage facility design process. The challenge of developing such an engineered tailings model is establishing more effective engagement and integration between disciplines (geology, geochemistry, mining engineering, metallurgical/process engineering, chemical engineering, mechanical/piping engineering, civil engineering, geotechnical engineering, dam construction (contractors), hydraulic engineering, environmental engineering, electrical engineering, hydrogeology, hydrology, chemistry, ecology, geography, economy, and sociology, among others). There is a need for experience and knowledge sharing [54,58].

## 7. Future Challenges in Reducing Socioenvironmental Impacts

Considering a more sustainable mining activity with respect to tailings governance, there are future challenges to reducing socio-environmental impacts. Changes must be carried out in the use of metallurgical technologies for mineral processing to generate less tailings, an environmentally friendly management of tailings reprocessing, and to establish permanent control of abandoned tailings deposits.

### 7.1. Reprocessing of Old TSFs (Abandoned TSFs): Is It a Sustainable Environmental Decision?

The copper mining industry mines ore at an average grade of 1.00% wt.% Cu. The metallurgic processing of ore results in tailings in an average grade range of 0.10–0.30 wt.% Cu (total copper) and 0.03 wt. % soluble Cu. Some copper mining companies are starting to recover copper by reprocessing tailings and are evaluating the reprocessing of other metals that currently may not have an economic value but may be attractive resources in the future [11,20]. Therefore, considering recent studies on old tailings storage facilities, it is possible to say that the deposited tailings material still contains around 0.3 wt.% Cu in some tailings storage facilities. Considering the current high prices of copper, together with the fact that the tailings material is already crushed and ground (representing 30% of the operating costs of mining), it can be considered as a potential georesource material [59] and is currently being studied worldwide for its potential exploitation [1]. Therefore, the reprocessing of many old mine tailings storage facilities may be a first-choice option for the remediation of an old mine site, providing a new opportunity for mining companies to relocate tailings material, reprocess it, and deposit it in an environmentally friendly way [1]. Typically, mining companies carry out the reprocessing of old TSFs using dredge pumps on a floating platform or by hydraulic mining monitors (Figure 26), and subsequently, slurry tailings are transported in pipelines to a processing plant.

In the mining industry, the circular economy is primarily related to the potential that exists in the reprocessing of tailings [60,61,62]. However, recovery of other elements and minerals (beyond critical metals) may require consideration of a holistic innovative approach. Some of the critical metals that were declared include cobalt (Co), vanadium (V), gallium (Ga), germanium (Ge), and rare earth elements (REE) [63,64]. Rare earth elements have 15 elements belonging to the group of lanthanides (lanthanum (La), neodymium (Nd), cerium (Ce), praseodymium (Pr), promethium (Pm), samarium (Sm), europium (Eu), gadolinium (Gd), terbium (Tb), dysprosium (Dy), holmium (Ho), erbium (Er), thulium (Tm), ytterbium (Yb), and lutetium (Lu)) plus yttrium (Y) and scandium (Sc) [42,65]. These minerals have multiple applications in which their use in the development of modern technologies, such as optics, lighting, LED screens, permanent magnets, and portable batteries stands out [66]. It is also used as a catalyst in petroleum refining processes, in the manufacture of ceramics, and in the military industry. Today smartphones, televisions, wind farms, low consumption light bulbs, hybrid vehicles, and optimal fibers are a reality thanks to these minerals. Of the total use of rare earth elements, 90% goes to electromobility and renewable energy generation. Electric cars carry 10 times more rare earth elements than conventional cars, and wind turbines are big consumers of the material [63,64].

However, there are environmental risks associated with the extraction and processing of REEs, and the magnitude of these risks is highly variable and depends on the nature of the deposit, the minerals from which the elements are extracted, the gangue minerals present in the deposit, and the toxicity of contaminants in mining waste. During extraction and processing, emissions to the air (fine particulate matter with possible content of metals and radioactive elements) and emissions of radioactive metals, acids, and turbidity to the water are produced. Another important issue is ammonium contamination: REE ion adsorption in situ leaching uses large amounts of ammonium sulfate, which remains in the wastewater [67]. It is estimated that to produce one ton of rare earth oxide by ionic adsorption, 300 m^2^ of vegetation and soil are removed, 2000 tons of tailings and 1000 tons of wastewater with high concentrations of ammonium and heavy metals are generated. When the ionic adsorption process is carried out in situ, the removal of vegetation is less, but the impacts of groundwater contamination, erosion, and mine closure are increased; in addition, the remediation of the site after mining activity becomes much more difficult due to aggregate contaminants [68].

Traditionally, due to experiences that happened throughout the world, it is known that mining is responsible for adverse effects that cause: (i) environmental damage, (ii) health problems, (iii) forced population displacement, (iv) increase in social inequality and (v) corruption, etc. [69]. In addition, considering the experiences of mining projects in China, the production of REE generates significant social and environmental impacts due to the use of strong and corrosive chemicals, and also due to the generation of radioactive co-products, such as uranium (U) and thorium (Th) [69]. However, if a comprehensive governance strategy is applied, managing the activity responsibly, the mining industry can generate positive social and economic benefits, such as job creation, better education, investment, innovation, and improvement of infrastructure [69].

Likewise, the list of critical metals changes with time as new manufacturing processes or new mines change the demand and supply situation. This could result in secondary products for mining companies that may choose to boost local markets by providing a regional entrepreneur with the opportunity to receive and manage material based on the exploitation of tailings, being a viable option for the development of economic, organizational, technological, environmental and social of the community. The authors expect that in the coming decades, many of the old tailings deposits could be reprocessed, depending on the future prices of metals, new uses for metals that are not currently economically valuable, and the application of more efficient metallurgical technologies.

The questions that must be asked are: How should reprocessed tailings be managed? Where should reprocessed tailings be deposited? Why are environmental management and metal recovery not combined? How can reprocessing of tailings be conducted without adversely affecting new areas?

In this context, in-pit disposal of tailings appears to be an attractive alternative for responsible reprocessing of tailings from closed mines. For some mines, this can signify a chance to apply a sustainable development action, with the objective of rehabilitating sites environmentally impacted by tailings. A large portion of the huge amount of deposited tailings should be removed for a number of reasons, such as: (i) relocation to a safe storage area, considering environmental risks; (ii) metallurgical reprocessing (extraction of heavy metals which can be toxic to environment); and (iii) relocation to ensure a stable mining closure (and minimize socioenvironmental liability).

### 7.2. Deep-Sea Tailings Disposal (DSTD): Is It a Stable Long-Term Solution? What Are the Environmental Impacts?

The deep-sea bed, until recently a remote and largely pristine environment, is now subject to growing anthropogenic pressures from industrial-scale resource extraction, accidental pollution, and deliberate waste disposal. Deep-sea tailings disposal (DSTD) involves the discharge of finely ground rock slurry from an outfall below the base of the surface mixed layer [70]. The tailings then flow as a near-bed density current to depths >1000 m (Figure 27), based on [71]. The technique builds on experience gained from more than a century of tailings disposal in Norwegian coastal fjords, in some cases to water depths of several hundred meters. The land use competition posed by tailings impoundments in central Chile makes DSTD an attractive and economic disposal option for some mining companies. It is currently used in Indonesia, Papua New Guinea, Greece, Norway, and Canada. The practice is highly controversial, with many local communities and non-governmental organizations voicing concerns about potential environmental impacts. DSTD entails massive inputs of fine sediment containing residual heavy metals derived from the terrestrial ore body (and potentially additional contaminants introduced by chemical processing of the ore) into bathyal environments regarded as hotspots of deep-sea biomass and biodiversity. The lack of information on its ecological consequences is, therefore, a significant gap in our knowledge of the anthropogenic impacts of mine tailings on the deep sea [10,11,12,13].

The following are commonly mentioned disadvantages and risks of DSTD [9,10,11,12,13]:Smothering of benthic organisms and physical and geochemical alteration of bottom habitat.Reduced numbers of species and biodiversity of marine communities.Risk of release of toxic elements from the tailings into the seawater.Bioaccumulation of heavy metals through food chains and ultimately into fish consumed by humans, with associated human health risks.Inability to recover deposited tailings (possible loss of valuable resources).Larger footprint on the seabed than on land and resuspension events.Sediment instability and slope failures caused by seismic and tsunami events.Potential toxicity of the flotation reagents used in the marine ecosystem (ecotoxicity).Plume sharing and dispersal of the fine particles throughout the sea.Relocation of tailings in different places of the marine ecosystem due to upwelling and currents.Near impossibility of controlling accidents.Low social acceptance.Reduced fishing activity.

### 7.3. Closure Phase of TSFs: Is It a Post-Operational Phase or a Perpetual Monitoring Phase?

Tailings are the most visible remaining signs of mining activity that, together with mine waste rocks and open pits, are recognized as the “legacy” impacts of mining (Figure 28). In the past, the primary aim was to provide a well-engineered structure into which tailings could be deposited without a great deal of attention given to closure requirements or long-term management of tailings storage facilities (TSF). Now, the closure of a TSF must be planned at the beginning of the project so that environmental, health, and safety impacts do not persist over time after the closure (perpetuity) [72].

One of the main components that persist after closure is TSFs, which host residues that contain minerals, metallurgical reagents, and water that can dissolve and transport contaminants to the soil, groundwater, and surface water. Furthermore, site conditions may cause hydraulic erosion and soil liquefaction. For these reasons, water management of TSFs on a space and time scale is an important issue in maintaining long-term physical, geochemical, and hydrological stability after closure (perpetuity) [72].

As a result of the intensive exploitation of minerals in the northern and central regions of the country, large volumes of massive mining waste were generated in the last 150 years. This is because, for every ton of ore extracted from a mine, approximately 97–99% is discarded as waste throughout the beneficiation process. One of the most voluminous mining wastes is tailing. According to the latest database published by SERNAGEOMIN [42], there are a total of 742 tailings deposits in Chile, in 10 of the 16 regions of the country, of which two are under construction, 104 are in operation, 463 are inactive, and 173 are abandoned [43].

## 8. Discussion: Suggested Strategies for Improvement Tailings Governance in Chile and Worldwide

There are many challenges in mining that must be faced both in Chile and around the world in order to obtain tailings governance with safe performances and world-class excellence. It is known that traditional tailings storage methodologies generate socio-environmental problems because they: (i) have footprints of very extensive areas of influence, (ii) are highly visible and impact the community, (iii) store large amounts of water, and (iv) generate emissions of particulate material (dust) in the air.

Avoiding these difficulties and the associated risks implies a rigorous commitment to tailings governance and also requires the application of the best available technologies (BATs) throughout the life cycle of the mining project. It is important to mention that, if these measures are not taken from an early stage in tailings governance, high environmental and social costs can be incurred in the long term, considering the closure and post-closure stages of the tailings deposits.

Therefore, TSFs must be designed, built, operated, closed, and rehabilitated to reach a level of risk that is acceptable to all stakeholders during the life cycle of the mining project (community, environment, authorities, and mining leaders). Risk management strategies must consider a systemic approach to mine tailings governance, which must take into account diverse viewpoints and the expectations of the communities in which mining companies operate. The main objective that a TSF must have is to safely store the tailings considering the solid material and the water during the operation, closure, and post-closure phases.

Risk management in mine tailings governance must have sufficient flexibility and adaptability to allow management in the face of changing events and circumstances. These events or changes could mean: (i) increases in tailings storage in the TSF, (ii) unforeseen expansions, (iii) commissioning of entirely new facilities, and/or (iv) implementation of new technologies in TSFs.

Considering the history and evolution of tailings governance in Chile, taking into account different space-time scenarios during the 20th century, important advances were made in technical and economic issues, highlighting relevant environmental advances in the last two decades. These advances are undoubtedly very valuable and represent a great effort by a series of professionals to keep mining tailings safe, controlled, and physically stable. Of course, there are still pending issues in the permanent application of a systemic, comprehensive, and holistic approach to mining tailings governance, which requires the formation of multidisciplinary teams that involve different professionals who can guarantee: physical, geochemical, and hydrological stability of mining tailings deposits in the design, construction, operation, closure, and post-closure phases. Technical, economic, risk, environmental, and social studies show that this led to better decisions recently that reduced the negative environmental impacts of mine tailings, contributed to anticipating potential damage, and mitigated or eliminated adverse effects.

For this reason, it is relevant to perform further studies on unknown natural sites, such as the sea floor, to obtain sufficient information to make responsible decisions and have an appropriate long-term vision for impacts and adverse effects on the human population and the environment, which are still not fully understood. Thus, it seems unreasonable to make a decision to place tailings in the sea floor without knowing for certain their consequences or effects in the marine ecosystem and in the coastal communities. An equally important concern is respect for agreements signed by Chile regarding the handling of waste in the ocean, namely, the London Convention and Protocol [74], which were backed by Chilean regulatory authorities at international meetings of countries seeking friendly management of waste in the ocean. Experiences described in this article represent a contribution and lessons learned that must be considered by regulators in Chile in taking a precautionary approach that is a more rigorous and relevant socioenvironmental assessment when they are faced with proposals for submarine deep-sea tailings disposal in waters of their jurisdiction. A decision based on a socioenvironmental approach must be taken by the authorities in order to permit, with clear restrictions and limitations, or ban submarine deep-sea tailings disposal techniques in Chile.

The current production of tailings in Chile due to copper mining is immense, approximately 800 million metric tons per year. Only pounds of precious metals are extracted for each metric ton of ore processed [24,42]. The main technology for obtaining concentrates of valuable metals, such as copper, and thus generating mining tailings (residue) from the processing of sulfide minerals on an industrial scale (more than 100,000 metric tons per day), is foamy or froth flotation [75]. Foam flotation for the beneficiation of copper-containing minerals is conducted using various chemical reagents, such as: (i) sodium ethyl xanthate, (ii) sodium cyanide, (iii) copper sulfate, (iv) pine oil tar, (v) fatty acid soaps, (vi) dithiophosphates, collectors, (vii) foaming agents, and (viii) lime, among others [76], which are held in an aqueous suspension with the help of aeration and controlled mixing. In froth flotation, the use of reagents that can act at the solid–liquid and liquid–gas interfaces is essential to successfully separate the mineralogical species of interest from the pulp [54,77].

However, the collector and foaming reagents currently used in mining processes have different characteristics in terms of toxicity and persistence. For example, sodium ethyl xanthate presents a high level of danger to humans and a medium level of toxicity for other living beings [78]. Other reagents, such as dithiophosphate-1404, methylisobutylcarbinol (MIBC), and Dowfroth 250 (a foaming agent), are considered toxic chemicals that pose various potential risks to human health and the environment [79]. Therefore, tailings may contain anthropogenic chemicals used in the metallurgical process (xanthates, organic compounds, corrosive chemical reagents, etc.) as well as high concentrations of heavy metals from natural sources (Cd, Cr, Mn, As, Pb, Zn, and Mo, among others). The hazards posed by these chemicals must be considered in carrying out safe tailings management. Therefore, from geochemical and toxicological perspectives, most tailings cannot be classified as inert or harmless materials, respectively, and must be stored with due care for the environment [54,80]. Mining and mineral processing wastes, particularly tailings, are generally not inert and must be isolated from interaction with the environment. Even in circumstances where waste material is chemically inert, the volume of waste may overload the capacity of ecosystems [81].

Many socioenvironmental problems associated with tailings management in Chile are related to the potential contamination of soil, water (surface and underground water), and air [82]. A relevant feature to consider is to avoid dust or fine particulate entrainments carried by local winds throughout the life of the TSF. Dust emission controls will play an important role as good design and proper implementation will provide the primary control mechanism for dust in accordance with regulatory air quality requirements. Some dust control alternatives to implement are: soil cover (borrow material), top soil/re vegetation cover, phytostabilization, and binder material or chemical agglomeration [53].

In recent years, there were important advances in tailings dewatering technologies, tailings storage facility infiltration control, and mining tailings management around the world. However, tailings spill accidents still occur, such as the Mount Polley case in Canada (2014), Fundao Samarco in Brazil (2015) Corrego do Feijao Brumandinho in Brazil (2019), and Jagersfontain in South Africa (2022) [16], mainly due to: (i) causes of human origin (dam failure, inadequate tailings/water management, inadequate construction, etc.) and (ii) natural causes (floods, earthquakes, debris flows, etc.) [83,84]. This indicates that for these mining wastes, there is not yet a degree of security that would guarantee their long-term stability, provide sufficient assurance of safety to neighboring communities, and protect the environment [85]. As a result, tailings transport and storage activities require careful control and responsible socioenvironmental management [54,81].

Considering the latest technological, scientific, and economic advances applied in mining, today it is possible to consider that a tailings material with 0.3% by weight of Cu can be considered a georesource and makes it possible to apply a circular economy. Just 30 years ago, it was not possible to consider this tailings material as a georesource. Therefore, it is possible to consider that as long as mining tailings have metal concentrations that are in a mineralogical form that is exploitable and that the concentrations are higher than the average concentration in the earth’s crust, it could be considered as a georesource today, or for future generations, considering that in the future metallurgical methods and technologies will be better and more efficient in mineral processing [1].

The TSF must include the construction of perimeter channels for guaranteed hydrological stability and rainfall water management; one for the right margin, and another for left margin of the valley. These channels are formed by ditches of trapezoidal section lined, such as concrete cloth, geoweb, or precast concrete, with average slopes of 0.5%, and using corrugated steel pipelines in lengths with slopes over 10%. Both channels collect coming from surface water runoffs, restricting their entrance to the tailings impoundment, and discharging these flows downstream of the TSF [53].

Likewise, mining tailings frequently face social controversy, due to concern about the proximity of the tailings deposits to populated centers, the potential impacts on the surrounding water, the use of territories with heritage and cultural significance or the emissions of particulate matter, among others [86].

For its part, infiltration of tailings deposits also constitutes an important challenge that the mining industry must address, due to its potential negative effects on the health and quality of life of the population, the environment, and the development of other economic activities, such as mining, agriculture, and livestock [86].

Currently, the major socio-environmental conflicts involve 47% of the current tailings production in Chile and projections indicate that due to the increase in the production of copper concentrate, the volumes of mining tailings will increase, forcing the expansion of the current deposits and the construction of new deposits. To this it is possible to add that around 50% of the future national copper resources are found in the central zone (IV to VI Region), an area that concentrates the highest population density (60% of the country’s population) and a large number of productive activities related to infrastructure, industry, agriculture, and viticulture [86].

All the aforementioned aspects show that, at present, the management of tailings deposits is a concern of great relevance for society in general. Mining companies must face not only the challenges of a technical nature related to the design, construction, operation, closure, and post-closure of their deposits, but also those manifested from the environment, for which it is necessary to establish, as a practical use, the development of comprehensive management models that consider all the necessary aspects to ensure the good performance of these mining tailings facilities, minimizing their risk of failure and favoring optimal operation in accordance with the economic, social, and environmental development of our society [86].

Finally, considering the Las Palmas tailings storage facility collapse, the lesson is that it is important to implement closure and post-closure plans that guarantee the physical, geochemical, and hydrological stability of abandoned tailings storage facilities in perpetuity. An interdisciplinary review of abandoned tailings storage facilities should be implemented as soon as possible, and physical, hydrological, and geochemical stabilization solutions should be put in place to prevent potential collapse and fatalities in the future. This will require dam safety reviews, periodic inspections, and monitoring plans to control tailings behavior in perpetuity and protect the environment, health, and people’s livelihoods [58,72].

## 9. Conclusions

This research studied the history of tailings governance in Chile from 1905 to 2022. This corresponds to a history of more than one hundred years that shows progress from simple methodologies to the implementation of the best available technologies (BATs). Failures, reforms, adoption of improved methods, and international experiences were exposed. The capacity of mining in Chile to: (i) design, build, and operate tailings deposits that resist earthquakes, (ii) produce tailings that require less use of water for their transport and disposal, (iii) and propose solutions that are more environmentally friendly to their environment, are the value and testimony of the efforts of different professionals, politicians, authorities, and the general public.

The practical experience of mining in Chile never stopped, it continues day by day in the present and will continue to do so in the future. In the governance of mining tailings, the adoption of best practices worldwide, continuous learning from failures, and a focus on innovation are promoted. The community and the authorities are proactive in prohibiting inappropriate techniques and methods, creating new regulations and implementing continuous improvement.

The historical facts reported in this article attest that mining activity can be carried out in a responsible and environmentally friendly manner. A new generation of dynamic and serious professionals can advance the search for sustainable mining practices that benefit society under comprehensive and holistic approaches. A continuous learning style, considering the best available technologies (BATs), and the audacity to act are the hallmarks of tailings governance in Chile. We recognize that the history of tailings governance in Chile offers lessons for other parts of the world. There is no doubt that there is great experience of years to share in tailings management with the world, but it is also recognized that there are still pending issues in social and environmental issues, as well as developing adequate planning for the closure of abandoned tailings deposits [33].

Considering this scenario and the alternative of implementing deep sea tailings disposal (DSTD), the 1972 Protocol to the Convention on the Prevention of Marine Pollution by Dumping of Wastes and Other Matter (known as the ‘‘London Protocol’’; [74]) stresses a “precautionary approach” that requires that “appropriate preventative measures are taken when there is reason to believe that wastes or other matter introduced into the marine environment are likely to cause harm even when there is no conclusive evidence to prove a causal relation between inputs and their effects.” In this context, regarding the effects of the discharge of mine tailings into the ocean, Chile’s experiences described in this article represent a contribution that should be considered by regulators to develop a precautionary approach that considers human and ecosystem health, a more rigorous framework of regulation, and an integral assessment of potential harms when considering proposals for DSTD into the ocean [81].

Impact assessments of DSTD were poorly managed, with approvals being given despite insufficient detail provided in environmental assessments. DSTD is considered to be an inexpensive way to dispose of large volumes of mine tailings; however, the definition of “cheap” did not previously include placing a value on the marine environment or local communities [10].

Due to the demand, critical metals (cobalt (Co), vanadium (V), gallium (Ga), germanium (Ge), and rare earth elements (REEs)) are set to expand worldwide over the coming decades, it is important to encourage the understanding, knowledge, and applicability of critical metal recovery from primary and secondary Chilean deposits, bringing new opportunities for commodities beyond copper. The mining of rare earth elements from reprocessed mine tailings needs to be analyzed while considering its benefits and environmental impacts to guarantee a circular economy and sustainability [60,61].

Mining is and will always be an activity with a high environmental and social impact if the current metallurgical mineral processing technologies are not modified and improved [54,58]. Therefore, considering current metallurgical mineral processing technologies, a sustainable mining approach may aim to optimize mineral processing to obtain valuable metals, which implies: (i) increasing economic benefits, (ii) decreasing the impact on the environment during the operation of the mining project, and (iii) the minimization of closure and remediation costs [1]. Considering the current low-grade copper mining deposits and their costs associated with processing (mining, crushing, grinding, and concentration), today it is possible to say that mining tailings, with their respective copper grades, can be considered a georesource [78,83]. Consequently, any material from mining tailings today can be considered as part of the mining deposit, considering a long-term view, as long as it represents an enrichment of metals compared to the average concentration of metals present in the earth’s crust [1,87].

There is thus potential for an integrated holistic approach to incorporate a greater level of sustainability into the tailings storage facility design process. The challenge of developing such an engineered tailings model is establishing a more effective engagement and integration between disciplines (geology, geochemistry, mining engineering, metallurgical/process engineering, chemical engineering, mechanical/piping engineering, civil engineering, geotechnical engineering, dam construction (contractors), hydraulic engineering, environmental engineering, electrical engineering, hydrogeology, hydrology, chemistry, ecology, geography, economy, and sociology, among others). There is a need for experience and knowledge sharing [54,58].

The implementation of Codelco’s Talabre TTD TSF project for the production of tailings of 400,000 mtpd will undoubtedly mark a relevant historical milestone in the application of thickened tailings disposal (TTD) technology, being a deposit of dewatered tailings of great magnitude never seen before.

Due to declining ore grades in mines that are currently operating, and which are part of the mining firms’ development projects, they must make greater efforts to extract increasing volumes of material to maintain their production levels or to grow in line with market demand, which will result in a proportional increase in the amount of waste that must be disposed of, either as mine waste rock material or in form of tailings. It is estimated that the current 800 million metric tons per year of production of tailings could nearly double by 2035 [44].

The design and construction of new tailings deposits and the expansion of existing tailings deposits is possibly the greatest environmental challenge facing the copper mining industry to guarantee its expansion and even its continuity in a wide area of the country, which includes several regions in the north and center of Chile. However, due to the large copper reserves present in the country, it is known that the exploitation of copper deposits will continue for many more decades, and the disposal of new quantities of mining tailings will be required. The environmental permits, the authorities and the community will demand for the construction, operation, and closure of these new tailings deposits in the north and center of Chile and the most advanced of the best available technologies (BATs) [88].

Finally, the evolution of mine tailings governance in Chile, current practices, and changes that could or may need to be made to improve practice are presented, as are future challenges of incorporating advances and lessons learned concerning the reduction in socioenvironmental impact and achieving more environmentally friendly solutions.

## Figures and Tables

**Figure 1 ijerph-19-13060-f001:**
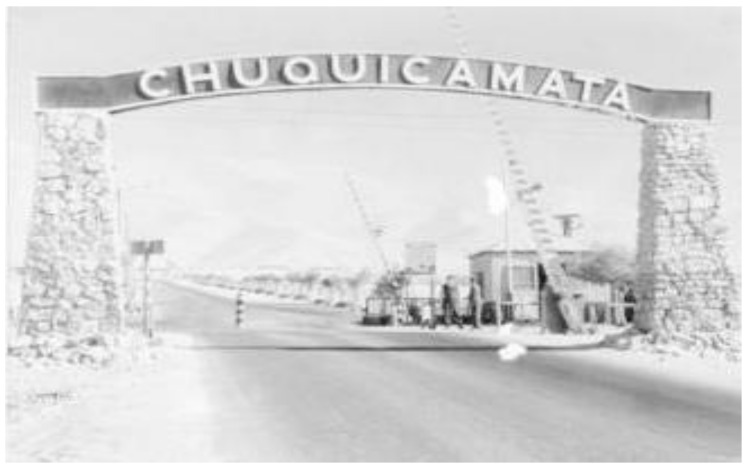
Chuquicamata Copper Mining Project [6].

**Figure 2 ijerph-19-13060-f002:**
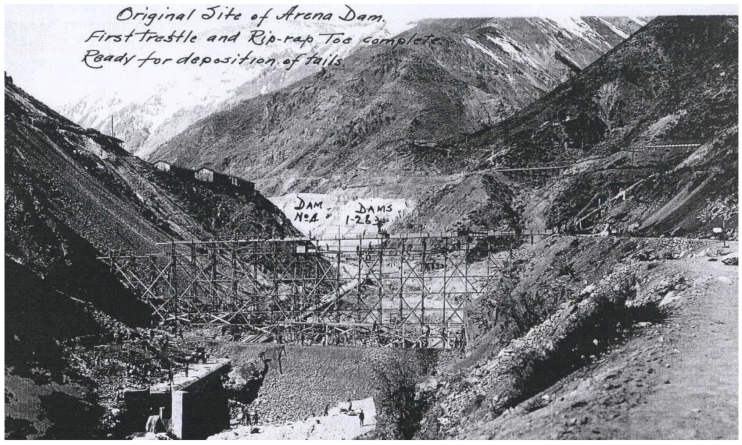
Arena Tailings Dam, Rancagua, El Teniente Mine, 1905 [15].

**Figure 3 ijerph-19-13060-f003:**
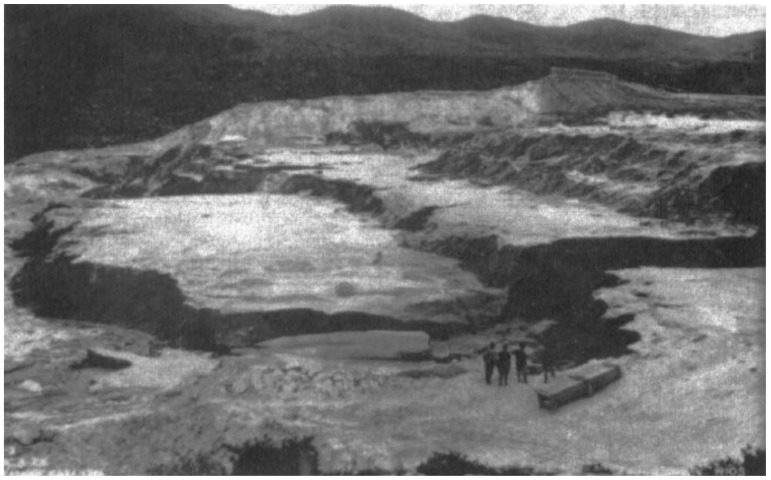
Collapse of Barahona Tailings Dam, Rancagua, El Teniente Mine, 1928 [16].

**Figure 4 ijerph-19-13060-f004:**
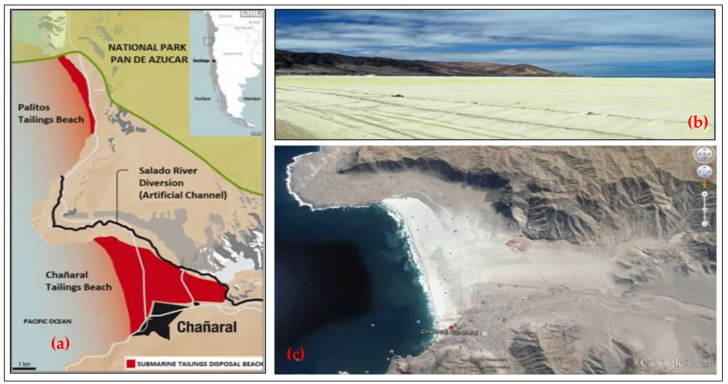
Off-shore tailings disposal at Chañaral Bay and Caleta Palitos. (**a**) Location. (**b**) Chañaral tailings beach. (**c**) Satellital image of the Chañaral bay [13,18,19,20].

**Figure 5 ijerph-19-13060-f005:**
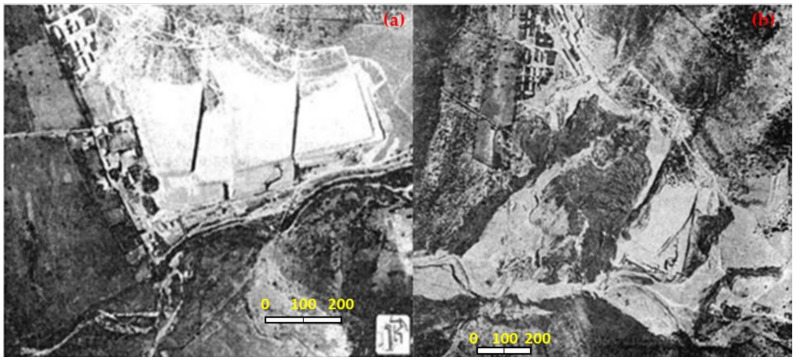
El Cobre Tailings Dam, Melon, El Cobre Mine, 1965. (**a**) Situation before the earthquake. (**b**) Situation after the earthquake [21].

**Figure 6 ijerph-19-13060-f006:**
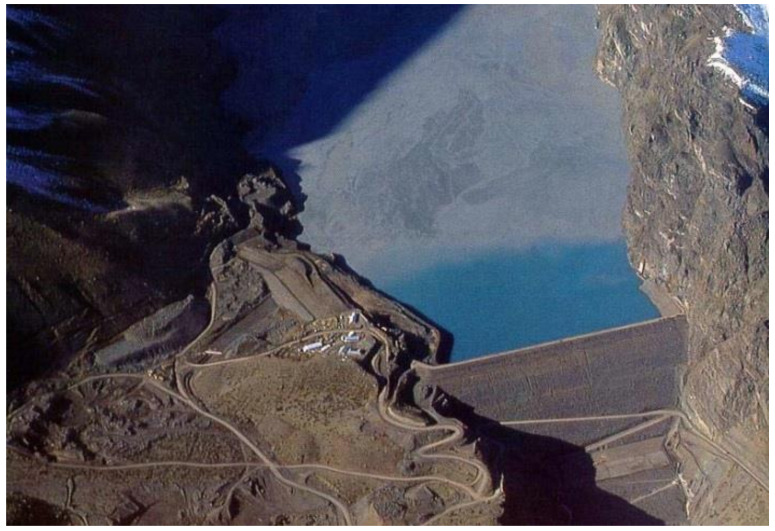
Los Leones Tailings Dam, Los Andes, Andina Mine [24].

**Figure 7 ijerph-19-13060-f007:**
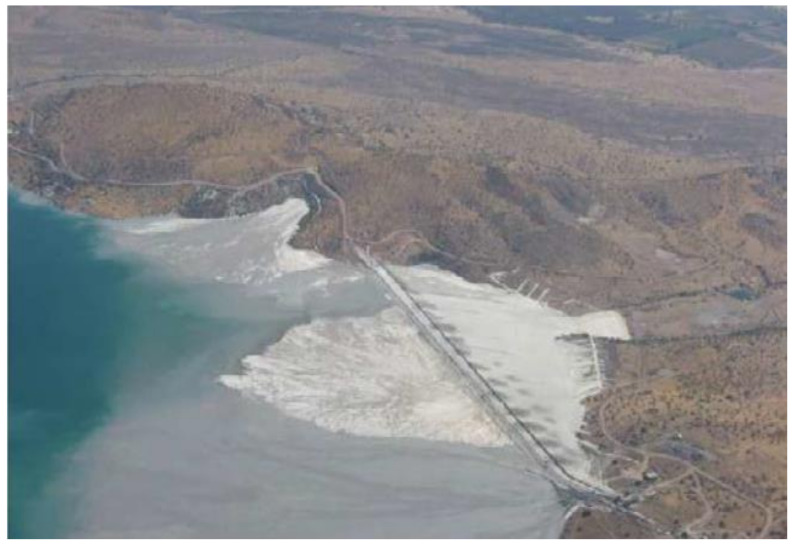
Las Tortolas Tailings Dam, Santiago, Los Bronces Mine, 2010 [25].

**Figure 8 ijerph-19-13060-f008:**
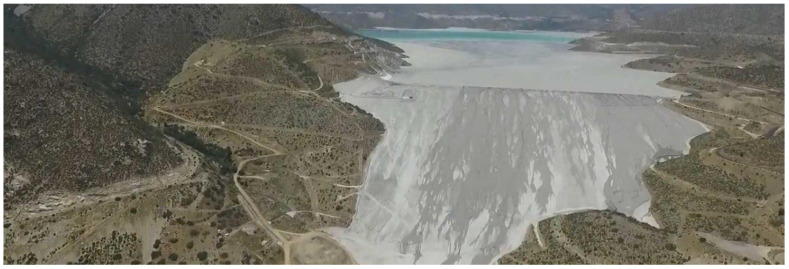
El Mauro Tailings Dam, Caimanes, Los Pelambres Mine [32].

**Figure 9 ijerph-19-13060-f009:**
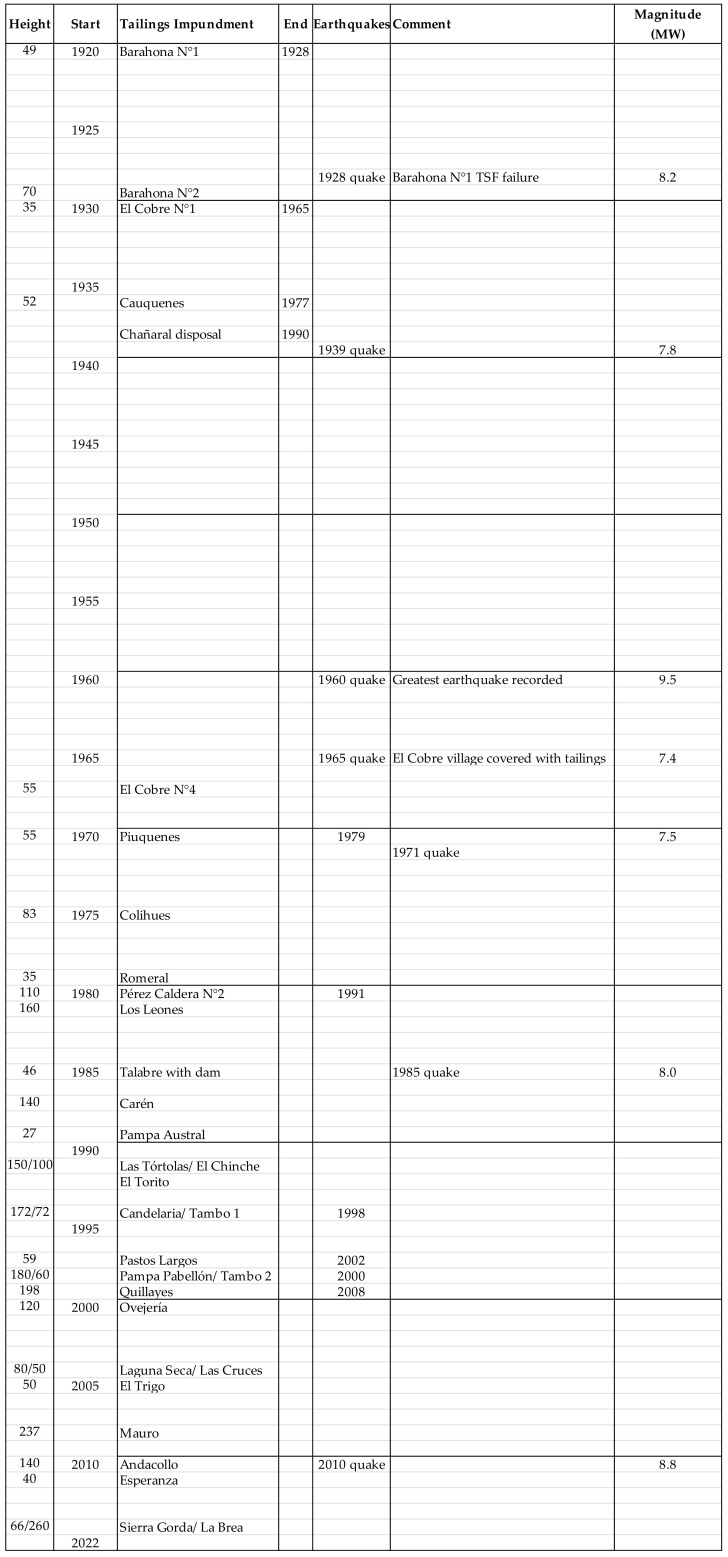
Timeline of the evolution of mine tailings management in Chile [33].

**Figure 10 ijerph-19-13060-f010:**
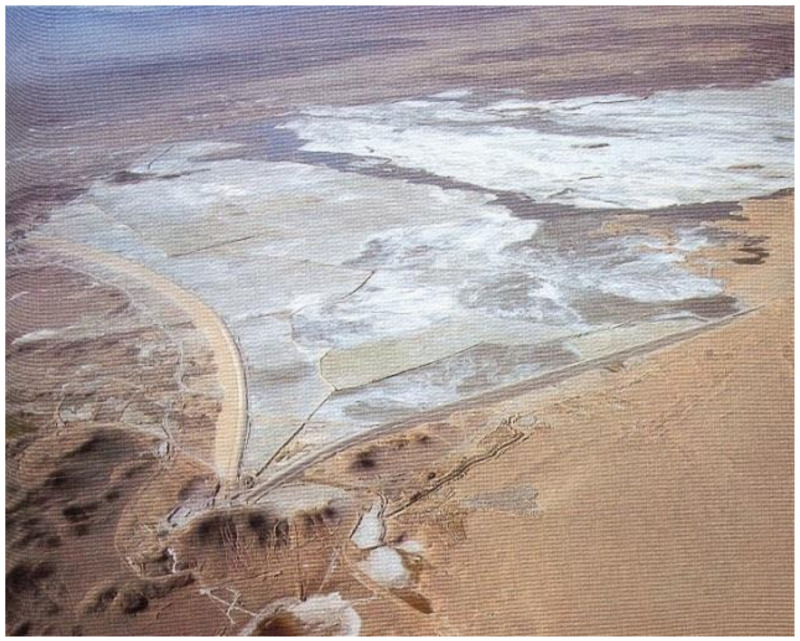
Talabre Tailings Dam, Calama, Chuquicamata Mine [34].

**Figure 11 ijerph-19-13060-f011:**
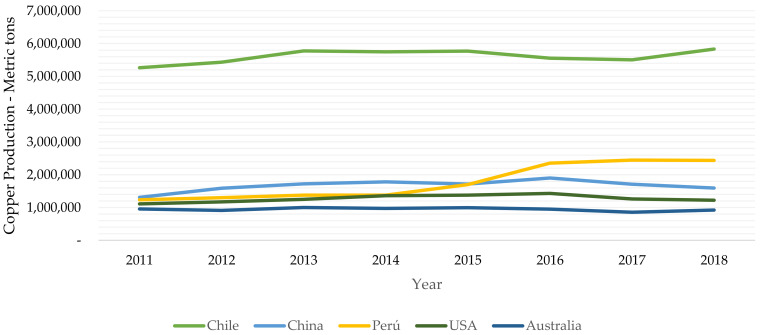
Copper production of the world’s leading copper-producing countries [7].

**Figure 12 ijerph-19-13060-f012:**
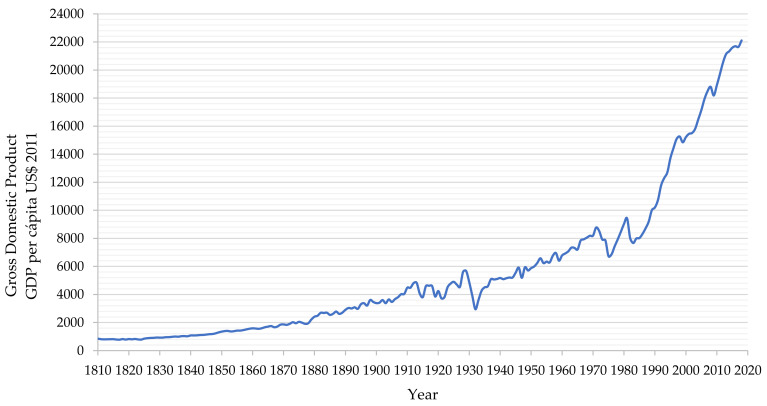
Historical evolution of gross domestic product per capita of Chile (1810–2018) [41].

**Figure 13 ijerph-19-13060-f013:**
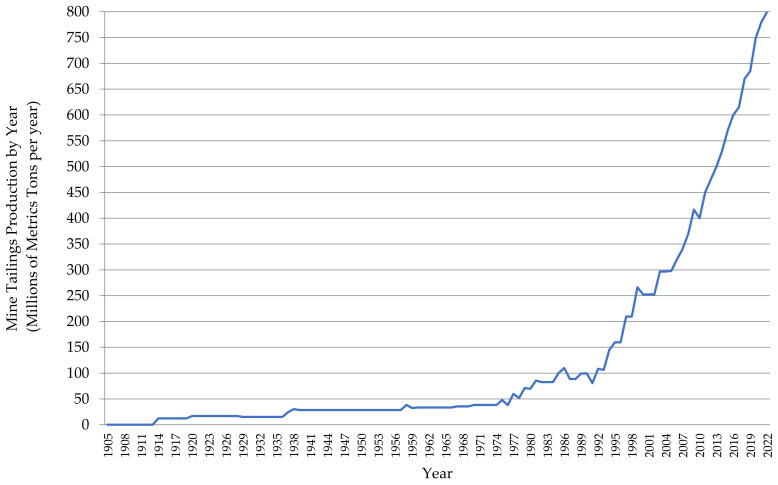
Historic mine tailings production per year in Chile—1905 to 2022 [24,42].

**Figure 14 ijerph-19-13060-f014:**
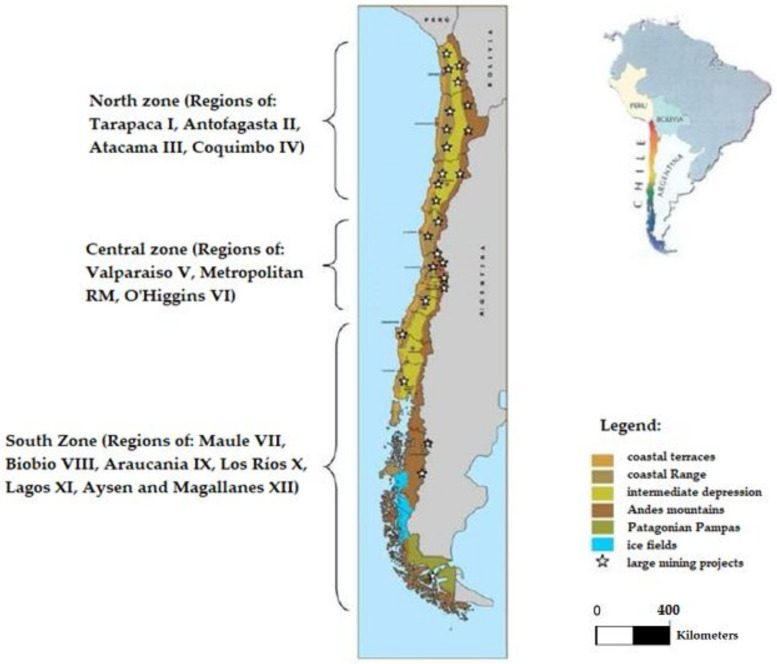
Map of Chile and location of large mining projects.

**Figure 15 ijerph-19-13060-f015:**
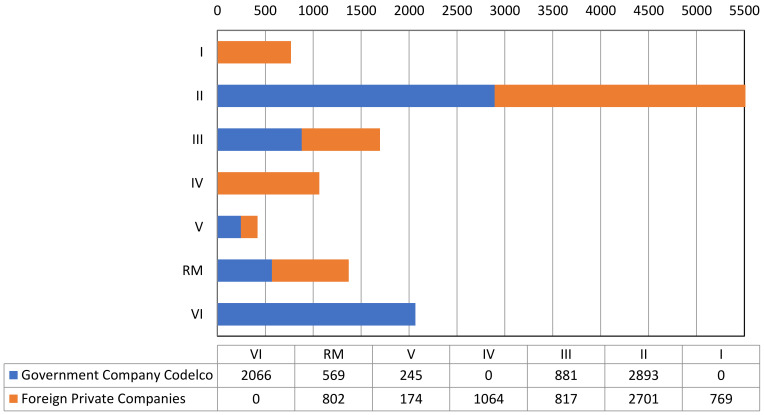
Tonnage amount of tailings stored in large tailings storage facilities per region in Chile 1905–2022 [24,42].

**Figure 16 ijerph-19-13060-f016:**
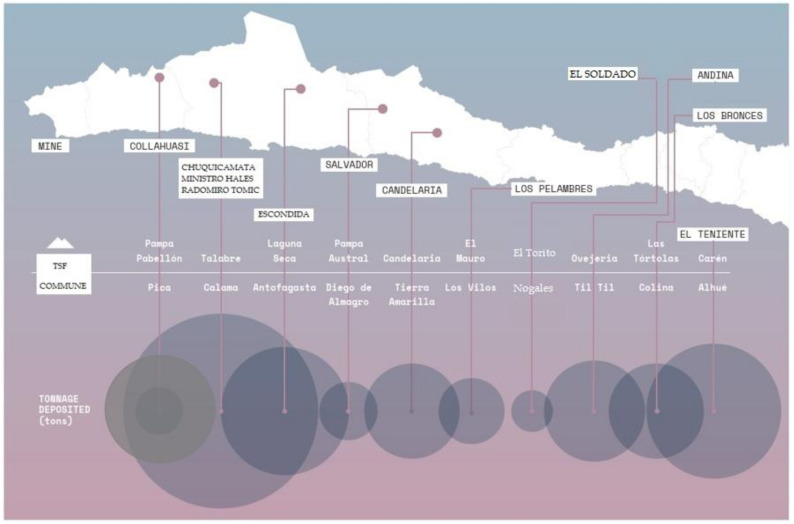
Top 10 Chilean tailings storage facilities in operation based on tonnage deposited (2019). Adapted from [24,42,43,44].

**Figure 17 ijerph-19-13060-f017:**
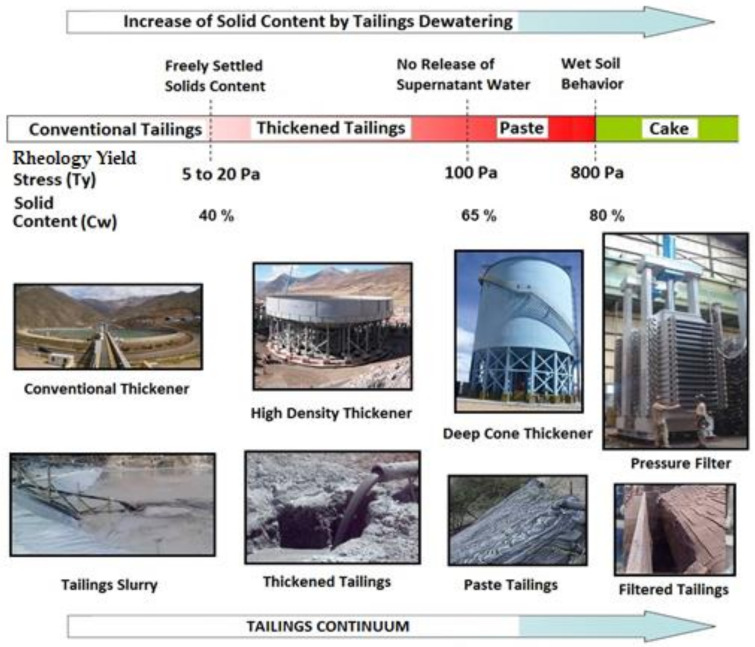
Dewatering tailings technologies–tailings dewatering continuum [47].

**Figure 18 ijerph-19-13060-f018:**
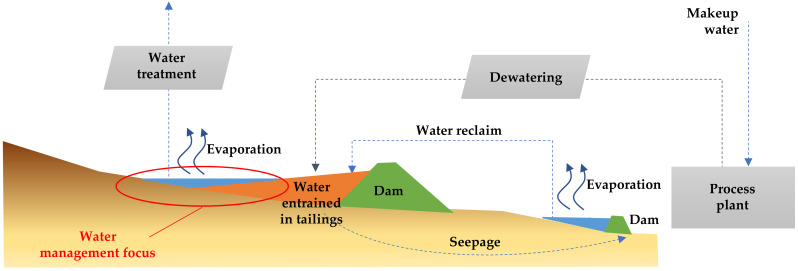
Schematic of a conventional tailings storage facility, adapted from [51].

**Figure 19 ijerph-19-13060-f019:**
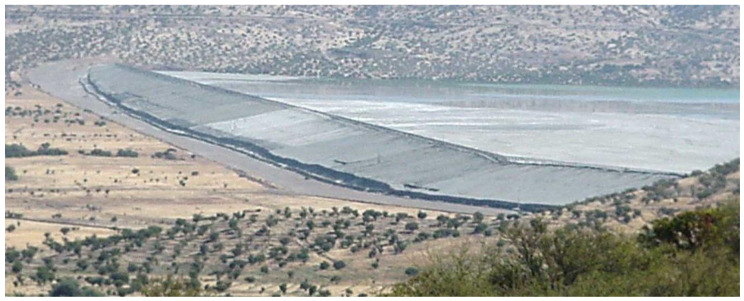
Ovejeria Tailings Dam, Andina Mine, conventional tailings technology [52].

**Figure 20 ijerph-19-13060-f020:**
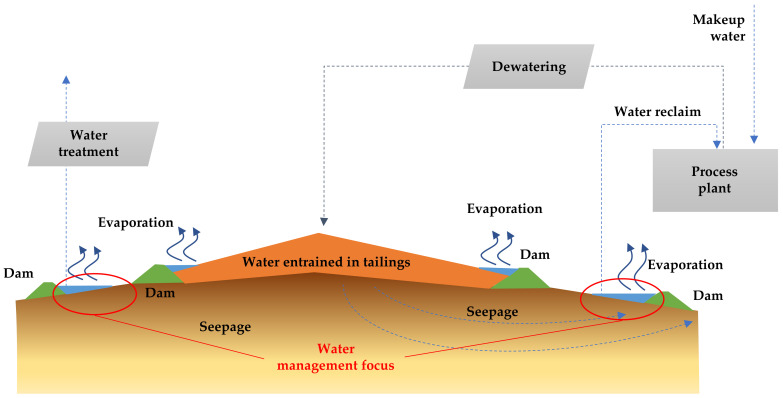
Schematic of a thickened tailings storage facility, adapted from [51].

**Figure 21 ijerph-19-13060-f021:**
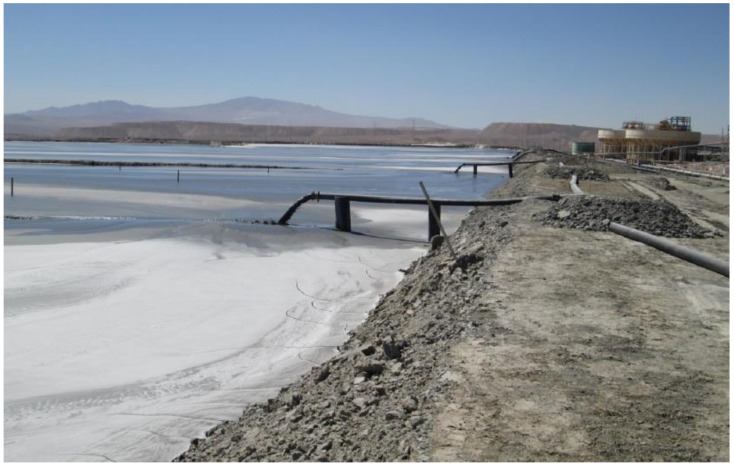
Centinela Tailings Storage Facility, Centinela Mine, thickened tailings technology [8].

**Figure 22 ijerph-19-13060-f022:**
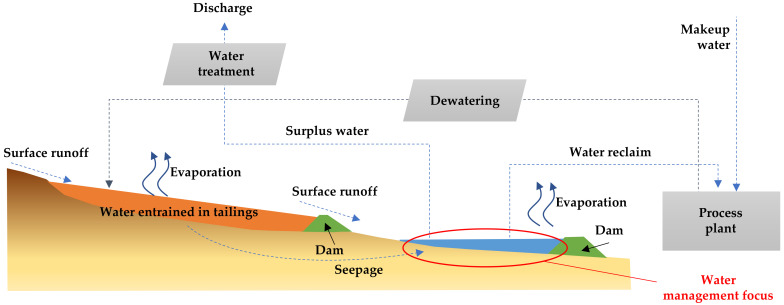
Schematic of a paste tailings storage facility, adapted from [51].

**Figure 23 ijerph-19-13060-f023:**
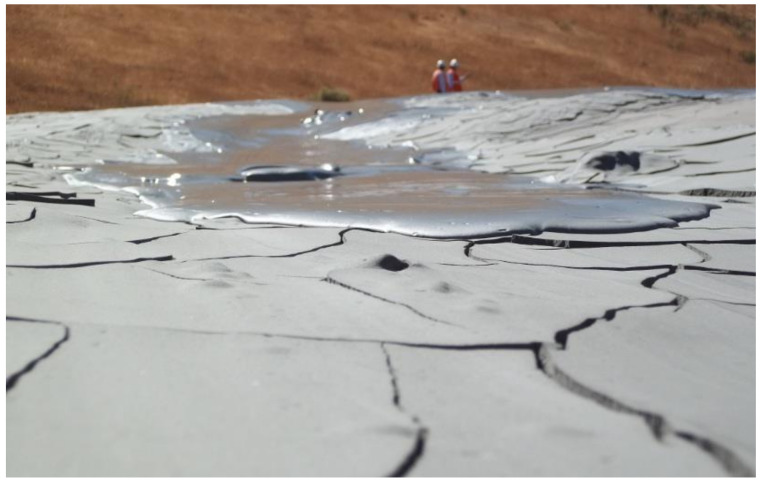
Chinchorro Tailings Storage Facility, Las Cenizas Mine, paste tailings technology [33].

**Figure 24 ijerph-19-13060-f024:**
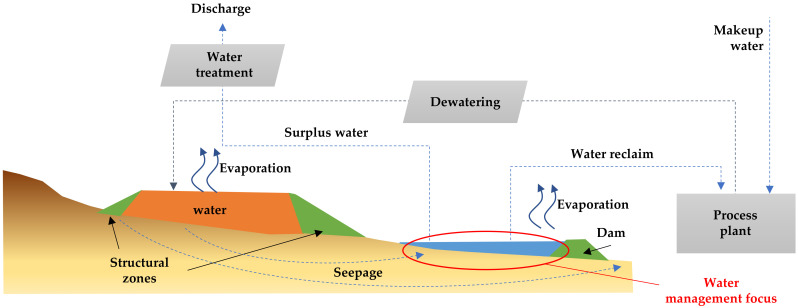
Schematic of a filtered tailings storage facility, adapted from [51].

**Figure 25 ijerph-19-13060-f025:**
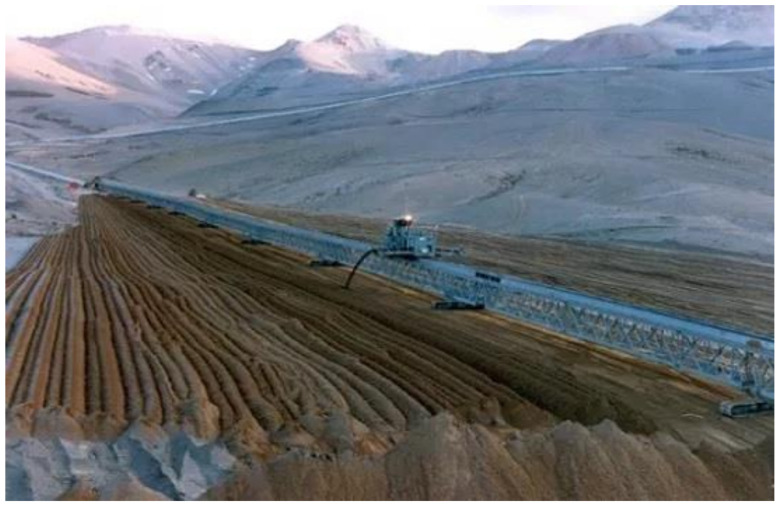
Rako Tailings Storage Facility, La Coipa Mine, filtered tailings technology [47].

**Figure 26 ijerph-19-13060-f026:**
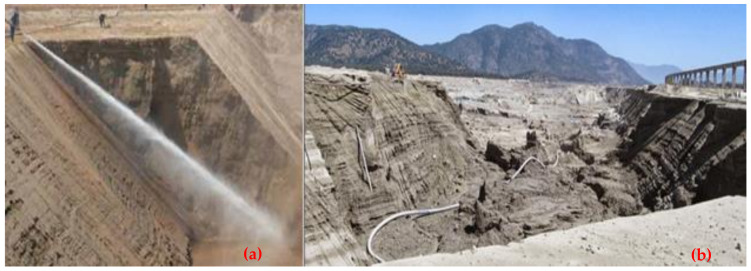
Tailings reprocessing by high-pressure water jets—Los Colihues TSF, El Teniente Mine. (**a**) High-pressure water jets operation. (**b**) Colihues TSF’s reprocessing [43].

**Figure 27 ijerph-19-13060-f027:**
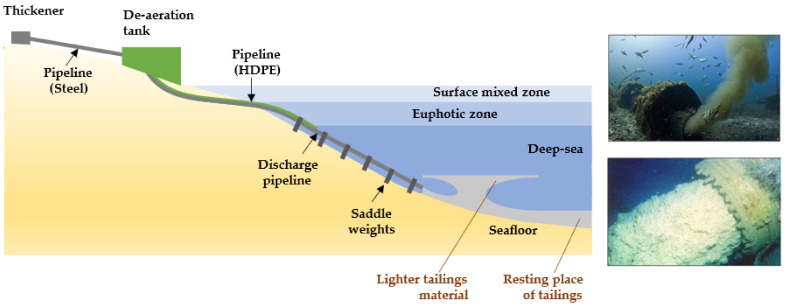
Idealized DSTP model for tailings pipe discharge plume. Adapted from [71].

**Figure 28 ijerph-19-13060-f028:**
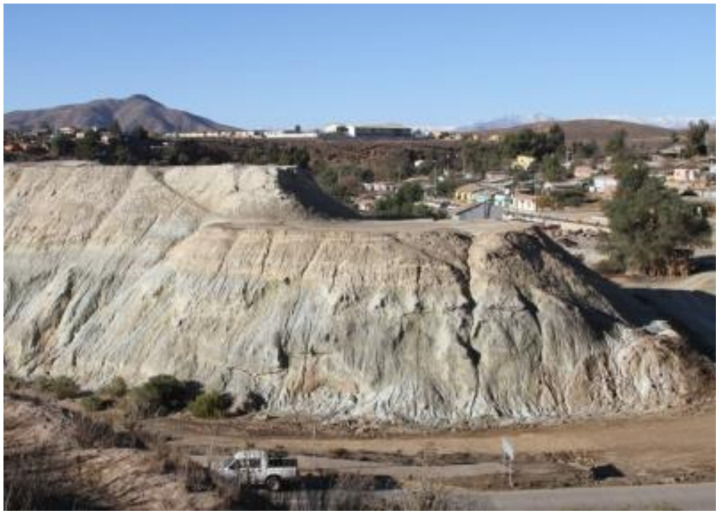
Abandoned tailings storage facilities in Andacollo City, Chile [73].

**Table 1 ijerph-19-13060-t001:** Statistics for total tailings storage for large-scale copper mining projects in Chile [24,42].

Region of Chile	Number of Large Tailings Storage Facilities	Mining Investments				Total Tailings Tonnage	Total Tailings Volume	Total Tailings Area
		Government Company (Codelco)	Millions of Metric Tons	Foreign Private Companies	Millions of Metric Tons	Millions of Metric Tons	Millions of m^3^	Ha
I	2	0	0	2	769	769	592	320
II	8	1	2893	7	2701	5594	3491	14,666
III	15	6	881	9	817	1738	1132	5896
IV	4	0	0	4	1064	1064	709	1468
V	4	2	245	2	174	419	279	620
RM	4	1	569	3	802	1371	914	648
VI	5	5	2066	0	0	2066	1377	3258
**Total**	**42**	**15**	**6654**	**27**	**6328**	**13,021**	**8494**	**26,876**

**Table 2 ijerph-19-13060-t002:** Current and authorized capacity for TSFs considering large-scale copper mining projects in Chile (2019). Adapted from [24,42,43,44].

Mining Operation	Region	Commune	TSF Name	Dam Material	Current Capacity (Tons)	Authorized Capacity (Tons)
Chuquicamata	II	Calama	Talabre	Cycloned tailings sand	1,792,722,179	2,060,420,000
Escondida	II	Antofagasta	Laguna Seca	Borrow material	1,302,238,493	4,500,000,000
El Teniente	VI	Alhue	Caren	Borrow material	1,275,600,044	3,288,000,000
Collahuasi	I	Pica	Pampa Pabellón	Mine waste rock	768,633,052	1,040,000,000
Los Bronces	RM	Colina	Las Tortolas	Cycloned tailings sand	678,320,902	1,000,000,000
Candelaria	III	Tierra Amarilla	Candelaria	Mine waste rock	538,906,120	563,561,793
Los Pelambres	IV	Los Vilos	El Mauro	Cycloned tailings sand	537,205,369	1,700,000,000
Andina	RM	Til Til	Ovejería	Cycloned tailings sand	482,847,856	1,930,000,000
Salvador	III	Diego de Almagro	Pampa Austral	Borrow material	303,140,000	305,396,837
El Soldado	V	Nogales	El Torito	Cycloned tailings sand	169,399,673	181,000,000

**Table 3 ijerph-19-13060-t003:** Characteristics of tailings storage facilities of copper mining projects in Chile [24,42,55].

Tailings Storage Facility Name	Mine Name	Tailings Technology	Tailings Production Rate (mtpd)	Dam Construction Material	Dam Height (m)
El Indio	El Indio	Conventional	Closure Phase	Borrow material	74
Piuquenes	Andina	Conventional	Closure Phase	Cycloned tailings sand	57
Los Leones	Andina	Conventional	Closure Phase	Borrow material	160
El Cobre No. 4	El Soldado	Conventional	Closure Phase	Cycloned tailings sand	55
Perez Caldera No. 1	Los Bronces	Conventional	Closure Phase	Cycloned tailings sand	90
Perez Caldera No. 2	Los Bronces	Conventional	Closure Phase	Cycloned tailings sand	145
Colihues	El Teniente	Conventional	Reprocessing 30,000	Borrow material	83
Cauquenes	El Teniente	Conventional	Reprocessing 30,000	Borrow material	35
Pampa Pabellon	Collahuasi	Conventional	170,000	Mine waste rock	90
Talabre	Chuquicamata	Conventional	200,000	Mine waste rock	50
Los Quillayes	Los Pelambres	Conventional	Closure Phase	Cycloned tailings sand	198
Mauro	Los Pelambres	Conventional	205,000	Cycloned tailings sand	237
El Torito	El Soldado	Conventional	20,000	Cycloned tailings sand	80
Ovejeria	Andina	Conventional	75,000	Cycloned tailings sand	130
Las Tortolas	Los Bronces	Conventional	125,000	Cycloned tailings sand	150
Pampa Austral	Salvador	Conventional	35,000	Borrow material	36
Caren	El Teniente	Conventional	180,000	Borrow material	70
Hamburgo	Escondida	Conventional	Closure Phase	Mine waste rock	25
Laguna Seca	Escondida	Conventional	370,000	Borrow material	50
Quebrada Blanca	Quebrada Blanca Phase II	Conventional	140,000	Cycloned tailings sand	310
Sand Stack	Caserones	Conventional	40,000	Cycloned tailings sand	500
Talabre TTD TSF Project (*)	Chuquicamata Ministro Hales Radomiro Tomic	Thickened	400,000	Mine waste rock	50
La Brea	Caserones	Thickened	50,000	Borrow material	248
Candelaria	Candelaria	Thickened	Closure Phase	Mine waste rock	160
Los Diques	Candelaria	Thickened	75,000	Mine waste rock	156
Andacollo	Carmen de Andacollo	Thickened	55,000	Mine waste rock	150
Esperanza	Centinela	Thickened	95,000	Mine waste rock	80
Catabela	Sierra Gorda	Thickened	110,000	Mine waste rock	75
Spence	Spence	Thickened	95,000	Mine waste rock	50
Cerro Negro Norte	Cerro Negro Norte	Thickened	20,000	Mine waste rock	73
Chinchorro	Las Cenizas	Paste	2500	Borrow material	30
Delta Plant	Enami	Paste	2000	Borrow material	20
Coemin	Coemin	Paste	8000	Borrow material	55
Alhue	Minera Florida	Paste	3000	Borrow material	25
La Coipa	La Coipa	Filtered	20,000	Borrow material	5
El Peñon	El Peñon	Filtered	3000	Borrow material	5
Coarse Tailings	Mantos Blancos	Filtered	12,000	Mine waste rock	5
El Gato	Atacama Kozan	Filtered	5500	Borrow material	5
Tambo de Oro	Punitaqui	Filtered	750	Borrow material	5
Tambillos	Minera Florida	Filtered	3000	Borrow material	5
Salares Norte	Salares Norte	Filtered	5500	Mine waste rock	5
Tailings Dry Stack	El Indio	Filtered	3000	Borrow material	5
Huasco	Huasco	Filtered	5000	Borrow material	5

Note: (*) Project under feasibility studies and environmental permits approval.

## Data Availability

The data presented in this study are available on request from the corresponding author.

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
