# Peer review of "Past, Present, and Future of Copper Mine Tailings Governance in Chile (1905–2022): A Review in One of the Leading Mining Countries in the World"

_ijerph, 2022, doi:10.3390/ijerph192013060_

Round 1

Reviewer 1 Report

This paper reviews the historical evolution and current practices of tailings management in Chile, and puts forward suggestions for future development. Overall, the paper is too long to be read and thus, the readability and logicality are decreased. Detail issues are given as follows:  

1. The text in the upper left corner of Figure 14 is not clear. Please modify it.  

2. Regarding the stability of tailings pond, Section 3 introduces the impact of earthquake on the stability of tailings pond. As we know, precipitation will also have a great impact on the stability of tailings pond.

3. In Section 5, please add a part of the content about the copper ore washing process introduced and the chemicals produced by the washing process, and how these chemicals affect the surrounding soil, water and other environment.

4. The instability of tailings pond will bring great results to the surrounding environment and human beings. This paper introduces a large number of tailings pond instability cases, and the stability monitoring of tailings pond is also the focus of attention.

5. Section 4.2 of the paper introduces the historical growth of the Chilean national economy and illustrates the importance of the mining industry to the Chilean national economy. Therefore, it is suggested to put this part into Section 4.1.

6. When the environmental pollution of tailings pond is mentioned in Section 5 of the paper, the pollution of metal elements to soil is mainly discussed. However, tailings dust is one of the main ways of tailings pollution, and the dust suppression method is also one of the important methods of tailings treatment.

7. Section 11 of the article gives the methods and ideas for improving tailings pond governance, and suggests adding an excellent case of tailings pond governance.

Reviewer 2 Report

GENERAL COMMENTS AND SUGGESTIONS

The article deals with a very current and interesting topic. The management and governance of waste mining material is a problem that must be addressed to make mining sustainable and green.

Despite my positive attitude, I believe that the manuscript should be supplemented as follows:

-       Despite this is a review article, the paper is too long! You provide many info that are not necessary to the aims of the paper (I highlight many example in the following).

-       The paper is also crowed with figures. I suggest preparing a single panel with the figures (for example, Figures 1-2-3 or Figures 6 to 9).

-       This is a review. I think that the number of references is insufficient. Much information lack of the correct references.

-       This paper is about the generic tailing in Chile, not only about the copper tailings. Sometimes this is not very clear.

-       Many concepts are repeated (I highlight many example in the following).

-       In general, the phrases are (very) too long. Please check careful the English construction of the sentence. I suggest using short word as possible.

-       The references! Check the instruction for the author and re-number the reference list.

-       The division in chapter is so confuse and, in some case, not necessary. I suggest a new chapter division:

o   Introduction: what is a tailing? Explain the characteristic, problem, and solution (generic not only in Chile).

o   Aims (short and clear!).

o   Chile mining activity (not only copper): past and present.

o   Chile tailing: specific characteristic, where are they from? Flotation etc. etc.

o   Past management (before the regulation) and disaster.

o   Present management (excursus on the regulation, in the chronological order).

o   Suggest for the future: holistic approaches etc. etc.

o   Conclusion (short and clear!).

-       In summary, you have to remember that the subject of the paper is the tailing management, not mining activity of copper deposits!

Title

I suggest shortening the title. As an example: “Past and presence of mining tailing management and governance in Chile: a review”.

Keyword

You have chosen composed word and very (too) long. In my opinion this not useful.

Chapter 1

Re-organize the chapter. The subject is the tailing! Not the copper deposit. Much info about copper is not necessary.

Chapter 2

-       I suggest changing the title into: “The evolution of tailing management and governance in Chile during the 20th century”.

-       2.1: change the title, the subject is the tailing not the mining! From line 105 to 118 this section is generic about mining activity not tailing management. Move to the introduction or in the specific section regarding the mining activity.

Chapter 3

-       Change title. This is the part about the law.

-       Change the title of the sections with the name of the law.

-       3.1 Add in this section the part in line 247 and following.

-       3.2 In this chapter you have to describe the regulation, not the disaster. Move the disaster in another chapter.

-       3.3 Explain better.

-       3.4 This is the generic introduction to the chapter 3. Move at the beginning.

Chapter 4

-       Transform this chapter in a generic introduction to the Chilean mining activity (if you want, with a little focus on copper).

-       Anyway, much info is useless. For example, from line 422 to 434. The subject need to be always the tailing!!

-       4.2 is it necessary?

Chapter 5

-       Delete this chapter. This is too general; a lot of information should be used as an introduction.

Chapter 6

-       Much information is generic, and you have to move in the introduction of tailing. In this section you have to provide specific example about Chile.

-       Create a specific paragraph for the water (from line 640 and following).

-       Together the photo, can you provide a scheme of the technique?

Chapter 7

-       Move this part in the chapter of the law.

Chapter 8

-       I do not understand if these are the suggested strategy or the strategy that the Chile has already adopted.

-       As already highlight, much info is too generic and not specific!

Chapter 9

-       Can you provide a short introduction at the beginning?

-       From line 980 to 1034, this is a mini-review of tailing reprocessing, this info is not useful here.

-       I think that the examples in section 9.4 and 9.5 do not help to clarify, but are confusing… Delete.

Chapter 10

-       Much info is already provided in other chapter, delate.

Chapter 11

-       These are the suggested strategies…change the title.

Conclusion

-       Short and clear! Re-write it according to the suggests providing for the paper.

Figures

-       Many figures have a very low resolution. In some cases, lack of the scale!

-       Add in the caption the year of the historical photo.

Figure 4: Explain all the three image and add A, B, C.

Figure 5: Can you improve the resolution of the figure? and describe each figure (A and B). Add a scale!

Figure 10: This is the most important figure of the paper! but it is unreadable! I suggest cancelling the photo of mines (already provide in the text).

Figures 11-12-13 If you choose to take all of these figures, you have to uniform the style. In Figure 11 the lines are too thick. In Figure 12, put the year in the vertical orientation (like figure 13), delate some of the lines. In Figure 13, the vertical lines are too much. Please decrease the spacing.

Figure 21 Each the picture are not clear, please provide the description of all the two figures.

Tables

Tables 2 Delete from all the name TSF to simplify the table.

SPECIFIC COMMENTS

Line 12 Change “the country” into “a country”.

Line 60 Delate mainly.

Line 62-65 Explain the differences.

Lines 99-102 This is a conclusion.

Line 104 (and following) Change “twentieth” to “20th”.

Line 113 “Over” lowercase letter.

Line 147 Change “homes” into “houses”.

Line 159 A “strange” dot after facility.

Line 164 Provide a map with the location of the mining area.

Lines 173-178 Too long sentence.

Line 180 give some examples of heavy metal and reagent.

Line 184 the 1990 is not early 20th century.

Line 196 “at the beginning” but the section is about the middle 20th century (1930-1970).

Line 199 Another “strange” dot.

Lines 206-211 too long sentence.

Lines 211-214 too long sentence.

Lines 219-228 Very interesting information, but it is not useful for the paper. The article is already very long, delate it!

Line 234 Another “strange” dot.

Lines 247-250 This is part of the regulation section (another chapter).

Lines 268-275 I do not understand this paragraph. I think that sentences are not necessary.

Line 330 In the text you use “cycloned tailings sand dam” very frequently. You may provide an abbreviation? Maybe: CTSD?

Line 354 Non-Newtonian lowercase letter.

Line 428 Another “strange” dot.

Lines 528-533 Nice...but it not necessary. Delate.

Lines 543-581 Simplify the text!

Line 879 “Acid rock drainage (ARD) prevention and mitigation” but not only…I hope.

Line 950 What does it mean “environmental passives”? “Old” need a lowercase letter.

Line 961 Another “strange” dot.

Line 966 Missing space after reference.

Lines 975-980 Use only the symbol of the elements.

Lines 1092-1105 Move to introduction.

Round 2

Reviewer 1 Report

All my questions have been resolved properly. But current state of the manuscript is still too long for a research paper. The readability is bad.

Reviewer 2 Report

Dear Authors,

I have read new version of your revised article. It is now much clearer and more readable. However, I still have a few little suggestions to improve the paper:

- The first paragraph of the introduction is still too long, I suggest starting by talking about the real topic of the article: copper tailing.

- Title: I suggest adding “Copper” into the title (e.g., Past, Present and Future of Copper Mine Tailings Governance in Chile (1905–2022): A Review in One of the Leading Mining Countries in the World)

- Figure 9: the figure is still not clearly readable…maybe the resolution has to be improved…

- Figures 12 and 13: To make the style of the figures as similar as possible, I suggest eliminating the vertical and secondary horizontal lines.

- Figure 14: Add the scale

- Figure 26a: it is unclear because of the resolution of the picture. Can you provide another picture?

- I think also the conclusion are still too long. Can you shorten them?

The number of figures refer to the new version.

Good Job! And Good Luck!
